# *SNTA1* gene rescues ion channel function and is antiarrhythmic in cardiomyocytes derived from induced pluripotent stem cells from muscular dystrophy patients

Eric N Jimenez-Vazquez[1], Michael Arad[2], Álvaro Macías[3], Maria L Vera-Pedrosa[3], Francisco Miguel Cruz[3], Lilian K Gutierrez[3], Ashley J Cuttitta[4], André Monteiro da Rocha[1], Todd J Herron[1], Daniela Ponce-Balbuena[1], Guadalupe Guerrero-Serna[1], Ofer Binah[5], Daniel E Michele[4], José Jalife[1,3,4]*

[1]Department of Internal Medicine and Molecular and Integrative Physiology, Center for Arrhythmia Research, University of Michigan, Ann Arbor, United States; [2]Leviev Heart Center, Sheba Medical Center, Tel Hashomer, and Tel Aviv University, Tel Aviv, Israel; [3]Centro Nacional de Investigaciones Cardiovasculares (CNIC) Carlos III, Centro de Investigación Biomédica en Red de Enfermedades Cardiovasculares (CIBERCV), Madrid, Spain; [4]Department of Molecular and Integrative Physiology, University of Michigan Medical School, Ann Arbor, United States; [5]Department of Physiology, Biophysics and Systems Biology, Ruth and Bruce Rappaport Faculty of Medicine, Technion - Israel Institute of Technology, Haifa, Israel

*For correspondence:
jjalife@cnic.es

## Abstract

**Background:** Patients with cardiomyopathy of Duchenne Muscular Dystrophy (DMD) are at risk of developing life-threatening arrhythmias, but the mechanisms are unknown. We aimed to determine the role of ion channels controlling cardiac excitability in the mechanisms of arrhythmias in DMD patients.

**Methods:** To test whether dystrophin mutations lead to defective cardiac $Na_V1.5$–Kir2.1 channelosomes and arrhythmias, we generated iPSC-CMs from two hemizygous DMD males, a heterozygous female, and two unrelated control males. We conducted studies including confocal microscopy, protein expression analysis, patch-clamping, non-viral piggy-bac gene expression, optical mapping and contractility assays.

**Results:** Two patients had abnormal ECGs with frequent runs of ventricular tachycardia. iPSC-CMs from all DMD patients showed abnormal action potential profiles, slowed conduction velocities, and reduced sodium ($I_{Na}$) and inward rectifier potassium ($I_{K1}$) currents. Membrane $Na_V1.5$ and Kir2.1 protein levels were reduced in hemizygous DMD iPSC-CMs but not in heterozygous iPSC-CMs. Remarkably, transfecting just one component of the dystrophin protein complex (α1-syntrophin) in hemizygous iPSC-CMs from one patient restored channelosome function, $I_{Na}$ and $I_{K1}$ densities, and action potential profile in single cells. In addition, α1-syntrophin expression restored impulse conduction and contractility and prevented reentrant arrhythmias in hiPSC-CM monolayers.

**Conclusions:** We provide the first demonstration that iPSC-CMs reprogrammed from skin fibroblasts of DMD patients with cardiomyopathy have a dysfunction of the $Na_V1.5$–Kir2.1 channelosome, with consequent reduction of cardiac excitability and conduction. Altogether, iPSC-CMs from patients with DMD cardiomyopathy have a $Na_V1.5$–Kir2.1 channelosome dysfunction, which can be rescued by the scaffolding protein α1-syntrophin to restore excitability and prevent arrhythmias.

**Funding:** Supported by National Institutes of Health R01 HL122352 grant; 'la Caixa' Banking Foundation (HR18-00304); Fundación La Marató TV3: *Ayudas a la investigación en enfermedades raras* 2020 (LA MARATO-2020); Instituto de Salud Carlos III/FEDER/FSE; Horizon 2020 - Research and Innovation Framework Programme GA-965286 to JJ; the CNIC is supported by the Instituto de Salud Carlos III (ISCIII), the Ministerio de Ciencia e Innovación (MCIN) and the Pro CNIC Foundation), and is a Severo Ochoa Center of Excellence (grant CEX2020-001041-S funded by MICIN/AEI/10.13039/501100011033). American Heart Association postdoctoral fellowship 19POST34380706s to JVEN. Israel Science Foundation to OB and MA [824/19]. Rappaport grant [01012020RI]; and Niedersachsen Foundation [ZN3452] to OB; US-Israel Binational Science Foundation (BSF) to OB and TH [2019039]; Dr. Bernard Lublin Donation to OB; and The Duchenne Parent Project Netherlands (DPPNL 2029771) to OB. National Institutes of Health R01 AR068428 to DM and US-Israel Binational Science Foundation Grant [2013032] to DM and OB.

## Editor's evaluation

This paper provides a novel correlation of cardiac with skeletal muscle phenotypes in muscle dystrophic disease. Pluripotent stem cells reprogrammed from cardiomyopathic Duchenne muscle dystrophic patients showed a dysfunctional $Na^+$ channel/$K^+$ channel channelosome. This correlates with a reduced cardiac excitability and conduction.

## Introduction

Null mutations in the Dp427 isoform of the dystrophin gene result Duchenne Muscular Dystrophy (DMD) (*Hoffman et al., 1987*). This inheritable X-linked disease affects primarily adolescent males causing progressive skeletal muscle deterioration, with negative effects in the central nervous system (*Anderson et al., 2002*). Muscular dystrophies are also characterized by cardiac muscle involvement (*Corrado et al., 2002*), which usually starts with an abnormal ECG (*Finsterer et al., 2018*). Eventually, most patients with DMD will develop cardiomyopathy by 20 years of age (*Shirokova and Niggli, 2013*). Many will be at a high risk for arrhythmia and sudden cardiac death (SCD), which contributes considerably to the morbidity and mortality of the disease (*Yilmaz and Sechtem, 2012*). However, diagnosis and prevention of arrhythmia are challenging in DMD patients (*Yilmaz et al., 2009*).

The mechanisms responsible for arrhythmias and SCD in patients with DMD cardiomyopathy are poorly understood. The dystrophin-associated protein complex (DAPC) is involved in mechanoprotection of the plasma membrane (*Petrof et al., 1993*). The DAPC acts also as a putative cellular signaling complex that forms a scaffold for numerous signaling and membrane ion channel proteins (*Constantin, 2014*; *Gavillet et al., 2006*; *Milstein et al., 2012*). The absence of dystrophin in DMD has the potential to alter trafficking, localization, and function of DAPC-associated proteins in skeletal and cardiac muscle (*Lohan et al., 2005*). For example, the expression and function of ion channels are defective in ventricular cardiomyocytes of the *mdx* mouse model (*Gavillet et al., 2006*; *Koenig et al., 2014*; *Rubi et al., 2017*; *Koenig et al., 2011*; *Albesa et al., 2011*). The absence of dystrophin in young *mdx* mice affects the function of $Na_V1.5$, leading to cardiac conduction defects (*Gavillet et al., 2006*). Inward rectifier potassium current $I_{K1}$ is reduced in the *mdx* mouse (*Rubi et al., 2017*) but the consequences of the disruption have not been identified.

Results from our laboratory and others strongly suggest that $Na_V1.5$ and Kir2.1 control cardiac excitability by mutually modulating each other's surface expression (*Milstein et al., 2012*; *Albesa et al., 2011*; *Petitprez et al., 2011*; *Leonoudakis et al., 2004*; *Matamoros et al., 2016*; *Ponce-Balbuena et al., 2018*). At the lateral membrane, $Na_V1.5$ and Kir2.1 channels form macromolecular complexes ('channelosomes') (*Pérez-Hernández et al., 2018*) that include α1-syntrophin, which is a part of the DAPC (*Gavillet et al., 2006*). Thus, we hypothesize that dystrophin gene mutations that truncate the Dp427 dystrophin isoform, disrupt $Na_V1.5$–α1-syntrophin–Kir2.1 interactions, altering the function of the most important ion channels controlling cardiac excitability and conduction velocity (CV), which would place the DMD patient at risk of arrhythmogenesis and SCD.

Here, we have used matured ventricular-like iPSC-CMs derived from two genetically distinct hemizygous DMD males, a heterozygous DMD female and two unrelated healthy subjects (controls) to investigate the mechanisms underlying the arrhythmias associated with loss-of-function dystrophin

mutations. We demonstrate that iPSC-CMs from patients with DMD cardiomyopathy have a dysfunction of the Na$_V$1.5–Kir2.1 channelosome, which leads to reduced excitability, slow conduction and reduced contractility. Importantly, DMD iPSC-CMs recapitulate the complex patterns of reentrant arrhythmias seen in patients. All such defects can be rescued by transfection with *SNTA1*, the gene coding the DAPC-related scaffolding protein α1-syntrophin.

## Methods

See Appendix 1 for details.

### Ethics statement

We obtained skin biopsies from two hemizygous DMD patients, one heterozygous female, and two healthy subjects after written informed consent, and consent to publish, in accordance with the Helsinki Committee for Experiments on Human Subjects at Sheba Medical Center, Ramat Gan, Israel (Approval number: 7603-09-SMC), and with IRB HUM00030934 approved by the University of Michigan Human IRB Committee. The use of iPS cells and iPSC-CMs was approved by the Human Pluripotent Stem Cell Research Oversight (HPSCRO #1,062 Renewal Approval, March, 2021) Committee of the University of Michigan, the Ethical Committee for Research at the Spanish National Center for Cardiovascular Research (CNIC), member of Carlos III Institute (CEI PI58_2019-v3), and the Regional Government of Madrid, Spain.

### Generation of iPSCs

Cell lines were generated using Sendai virus CytoTune-iPS 2.0 Sendai reprogramming kit (Thermo Fisher) for transfection of Yamanaka's factors, as described (*Eisen et al., 2018*; *Eisen et al., 2019*).

### Patient-specific iPSC-CMs monolayers (adapted from *Herron et al., 2016*)

We obtained highly purified iPSC-CMs after directed cardiac differentiation. After 30 days in culture, cardiomyocytes were purified, dissociated, and plated on Matrigel-coated polydimethylsiloxane (PDMS) membranes at a density of ~200 K cells per monolayer. Cells were maintained for 7 days before replating onto Matrigel-coated micropatterned PDMS for patch-clamp and immunostaining experiments. At least three separate cardiomyocyte differentiations were used for all the experiments.

### Micropatterning on PDMS (adapted from *Kuo et al., 2012*)

Stamps were sonicated and then incubated with Matrigel diluted in water (Corning, 100 µg/ml) for 1 hr. Then, 18 mm PDMS circles were UVO treated before micropatterning. An hour later, the Matrigel solution from the PDMS stamps was aspirated and each stamp was inverted onto each PDMS circle and removed one by one. The micropatterned PDMS was incubated overnight with pluronic-F127 at room temperature. Then, it was cleaned with antibiotic–antimycotic solution and exposed to UV light before replating cells. About 30,000 human iPSC-CMs were placed in the center of the micropatterned area. Cells were cultured on micropatterns at least 4 days prior to experiments.

### Electrophysiology

We used standard patch-clamp recording techniques to measure the action potentials (APs), as well as sodium current ($I_{Na}$), L-type calcium current ($I_{CaL}$), and inward rectifier potassium current ($I_{K1}$) in the whole-cell configuration. All experiments were conducted at room temperature, except for the AP recordings, which were obtained at 37°C and paced at 1 and 2 Hz.

### RT-PCR

For quantitative evaluation of mRNA expression in each experimental group, total RNA was prepared using the RNeasy Mini Kit (Qiagen), including DNAse treatment. cDNA was synthetized using Super-Script III First-Strand Synthesis System (Invitrogen). Quantitative PCR was performed using TaqMan Universal PCR Master Mix (Applied Biosystems) in the presence of primers for *SCN5A*, *CACNA1C*, and *KCNJ2*. We calculated mRNA fold expression by the ΔΔCT method using the 18S rRNA as the

housekeeping gene. Every qPCR reaction was performed in triplicate and repeated using cDNA from at least three separate cardiomyocyte differentiation cultures.

### Western blotting

Standard western blotting was applied and Image Lab software (BioRad) was used for analysis. Total and biotinylated protein was obtained from iPSC-CM monolayers and resolved on sodium dodecyl–sulfate polyacrylamide gel electrophoresis gels. Membranes were probed with antihuman dystrophin, $Na_V1.5$, and Kir2.1 antibodies, using Actinin as the loading control for total protein analysis, Na/K--ATPase for biotinylation experiments, and cTnT as the marker for cardiomyocytes.

### Immunofluorescence

iPSC-CMs were plated on micropatterned PDMS, fixed, treated, and analyzed as described in detail in Appendix 1s (see also *Herron et al., 2016*). Images were recorded with a Nikon A1R confocal microscope (Nikon Instruments Inc) and Leica SP8 confocal microscope (Leica Microsystems).

### Optical mapping

Optical action potentials were recorded from control and patient-specific iPSC-CMs using the voltage-sensitive fluorescent dye FluoVolt (F10488; Thermo Scientific). Unless otherwise indicated, we paced the monolayers using 3 ms pulses of 7–15 V at 1 Hz. Activation patterns were recorded, CV and optical action potential durations (APDs) were measured, and arrhythmia inducibility was determined as described previously (*Herron et al., 2016*; *da Rocha et al., 2017*). Briefly, we used a train of 10 pulses at 5 Hz followed by 10 pulses at 10 Hz. Trains were stopped once arrhythmia was induced.

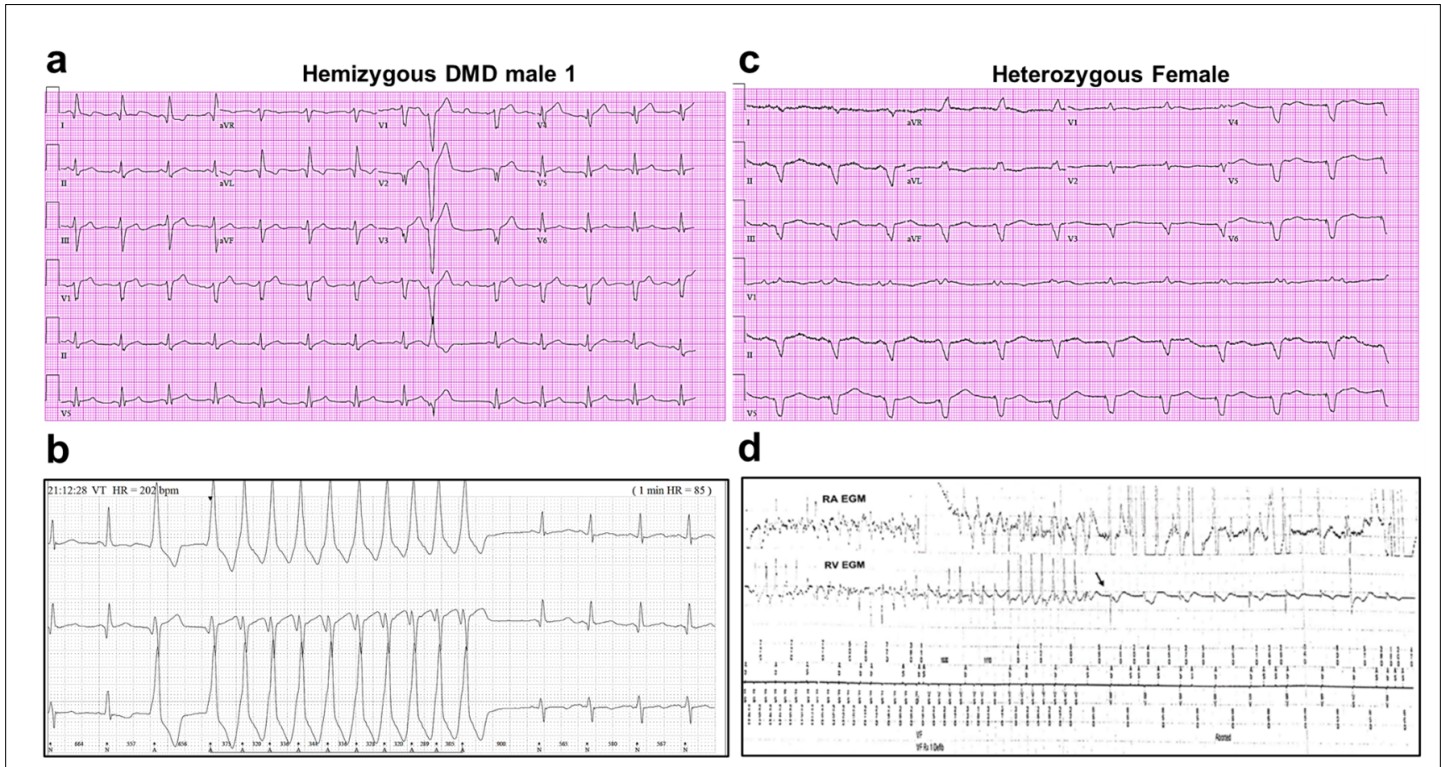

**Figure 1.** Altered ECG and arrhythmias in Duchenne Muscular Dystrophy (DMD) patients with cardiomyopathy. (**a**) Abnormal ECG in a 34-year-old DMD male: PR interval, 116 ms; QRS, 120 ms; QT/QTc, 404/472 ms; and PRT axes, 18-16-90. (**b**) Holter recording from the same patient shows nonsustained monomorphic ventricular tachycardia. (**c**) Abnormal ECG from the heterozygous female at 50 years of age: left axis deviation; QRS, 178 ms; QT/QTc, 564/612 ms; and PRT axes, 55-263-85. (**d**) Holter atrial electrograms of the heterozygous female shows atrial fibrillation with complete AV block after AV nodal ablation. Ventricular electrogram shows polymorphic ventricular tachycardia with spontaneous termination (arrow) and resumption of ventricular pacing.

## Generation and stable transfection of *SNTA1-IRES-GFP*

Nonviral piggy-bac vector encoding SNTA1-IRES-GFP was cotransfected with mouse transposase-expression vector into iPSCs. After 3–5 days GFP-positive cells were selected by FACS sorter and grow-up. Every week, fluorescence was confirmed, and cells sorted to confirm cDNA stable integration into the cells. After that, iPSC-CMs differentiation protocol was applied as stated above.

## Statistics

All data are expressed as mean ± standard error of the mean. In each dataset, a Grubbs' test was performed after data collection to determine whether a value should be considered as a significant outlier from the rest. Nonparametric Mann–Whitney test was used. Multiple comparisons were tested using two-way analysis of variance followed by *Sidak's* or Dunnett's test using Prism 8. $p < 0.05$ was considered significant. All experiments were performed as a single-blind study to avoid sources of bias.

# Results

## Clinical characteristics

We generated iPSC-CM lines from reprogrammed skin fibroblasts that were collected from three patients suffering from DMD cardiomyopathy. Two hemizygous males had a clinical and genetic diagnosis for DMD; the third patient was a DMD heterozygous female (*Figure 1*). iPSC-CMs from a healthy young male unrelated to the patients (Control 1) and iPSC CMs derived from a commercial iPS cell line (Product ID: CRL-2522) generated from normal foreskin from a neonatal male (Control 2) acted as negative controls. Complete clinical data were accessible for one DMD male and the heterozygous female. The hemizygous male (Male 1) harboring a nonsense point mutation in the dystrophin gene (exon 41) experienced DMD from early childhood, being diagnosed with dilated cardiomyopathy at age 17. Eight years later he was hospitalized in respiratory and heart failure (LVEF = 15%), requiring tracheostomy and prolonged ventilation. An ECG exhibited sinus rhythm with a narrow QRS and QR pattern in L1, AVL, and QS leads V2–3 (*Figure 1a*). At age 30, the patient became respirator dependent with a reasonably controlled heart failure. A routine Holter-ECG obtained 3 years later showed frequent premature ventricular complexes and episodes of nonsustained ventricular tachycardia at rates of up to 200/min. An ICD was implanted, which discharged appropriately 2 years later for repeated episodes of ventricular flutter deteriorating into ventricular fibrillation (*Figure 1b*). Three years later, the patient expired of heart failure at age 38.

The female patient, heterozygous for a deletion of five exons (Δ8–12) in the dystrophin gene, presented proximal muscle weakness with creatine kinase elevation at age 42. She had a son with DMD who died at 16. At presentation, she exhibited biventricular dysfunction with left ventricular dimension of 65 mm, LVEF of 30% and moderate-to-severe mitral insufficiency. At age 49, she developed severe biventricular dysfunction with LVEF = 20% and severe tricuspid regurgitation. She was in NYHA IV, and the cardiorespiratory exercise test showed a $VO_2$ max of 6 ml/kg/min, indicating a severely reduced aerobic capacity. An ECG obtained at age 50 revealed severe QRS widening and QT prolongation (*Figure 1c*). At that time, she had LVEF 30–35% and her heart failure was relatively well controlled. A year later, she developed paroxysmal atrial fibrillation with rapid ventricular response and recurrent episodes of nonsustained ventricular tachycardia (*Figure 1d*). AV nodal ablation and CRTD pacemaker-defibrillator implantation were required. The patient died at 51 in end-stage heart failure associated with renal insufficiency.

The additional DMD patient (Male 2) was a 13-year-old male carrying a 6-exon dystrophin deletion (Δ45–50). The patient was nonambulatory (used a motorized wheelchair) but respirator free at the time of the skin biopsy. He did not have significant cardiomyopathy at the time of collection, which was not surprising given his young age and the typical presentation of DMD cardiomyopathy as later onset (*de Souza et al., 2020*; *Szabo et al., 2021*). No follow-up information is available for this patient. The unrelated healthy individuals (Controls 1 and 2) have no personal or family history of DMD or any related disease.

## Dystrophin is absent in iPSC-CMs derived from hemizygous DMD patients

Compared to Control-1 iPSC-CMs and to left ventricle samples from a patient with Becker dystrophy, iPSC-CMs from hemizygous males were deficient in the full-length adult DP427 dystrophin isoform (*Figure 2a,b*, *Figure 2—source data 1*). The iPSC line named Male 2 shows a deletion of exons 45–50, while the other dystrophic cell line (Male 1) presents a nonsense point mutation (R1967X) in exon 41 of the dystrophin gene constituting a premature stop codon. The cell line generated from the 50-year-old DMD heterozygous female carried a deletion of exons 8–12. Notably, her iPSC-CMs showed expression of dystrophin protein like the control.

## Micropatterning controls cell shape and facilitates electrophysiological recordings

Cell shape is critical for cardiomyocyte electrical, mechanical, and contractile function (*Kuo et al., 2012*). Adopting the typical cylindrical morphology helps improve contractility, which promotes electrophysiological phenotype maturation (*Ribeiro et al., 2015*). When cultured on a nonmicropatterned smooth surface, DMD iPSC-CMs are flat shaped and have a frail membrane making them a challenge for patch-clamp experiments (*Figure 2c*, *left*). Therefore, we plated our iPSC-CMs on Matrigel-coated micropatterned PDMS (*Figure 2c*, *right*). The approach produces large numbers of thick cylindrical-shaped, binucleated cardiomyocytes with well-organized sarcomeres (*Figure 2d*), which are two important signs of maturation. Micropatterned iPSC-CMs are easier to patch. They are electrically excitable and their electrical phenotype approaches the adult human cardiomyocyte, with maximum diastolic potentials (MDP) of −70 to −80 mV, and APDs of 200–300 ms (see below; *Taggart et al., 1996*; *Grandi et al., 2011*). On the other hand, as shown in *Figure 2e*, unlike control cells, immuno-stained DMD cells do not express dystrophin, whereas iPSC-CMs from the female patient show variable expression of dystrophin.

## Action potentials in dystrophic iPSC-CMs have a reduced maximum upstroke velocity

Clinically, DMD patients may experience cardiac complications and often exhibit electrical conduction abnormalities and life-threatening arrhythmias (see *Figure 1*; *Fayssoil et al., 2010*; *Finsterer and Stöllberger, 2003*) At the cellular level, such alterations are often the result of reduced excitability. We therefore conducted patch-clamp recordings in micropatterned iPSC-CMs in the current-clamp configuration. In *Supplementary files 1 and 3*, we present comparisons at two different frequencies for DMD versus Control 1 (*Supplementary file 1*), DMD versus Control 2 (*Supplementary file 2*), and Control 1 versus Control 2 (*Supplementary file 3*). We quantitated AP parameters such as maximal upstroke velocity ($dV/dt_{max}$), overshoot, AP amplitude, MDP, and AP duration (*Figure 3*). Statistical analysis demonstrated that Controls 1 and 2 were very similar to each other, both exhibiting well-polarized MDPs, $dV/dt_{max}$ larger than 40 V/s and amplitudes larger than 100 mV. However, they both differed significantly from all three DMD groups (*Figure 3—source data 1*; and *Figure 3—figure supplement 1a-f*), particularly in terms of $dV/dt_{max}$. iPSC-CMs from both DMD male and female patients revealed abnormal AP profiles compared to both controls. For example, overshoot and amplitude were lower in the Male 2 cells compared to the controls. In addition, female DMD cells showed a more depolarized MDP than control iPSC-CMs (*Figure 3e*). Finally, no significant differences existed in $APD_{90}$ values and similar action potential parameter changes were obtained at 2 Hz (*Supplementary file 1*).

## Conduction velocity is impaired in DMD iPSC-CM monolayers

The reduced $dV/dt_{max}$ at the single-cell level suggested that CV may be compromised in iPSC-CMs monolayers from affected individuals. Hence, we conducted optical mapping experiments using the voltage-sensitive fluorescent dye FluoVolt in control, DMD, and female iPSC-CM monolayers paced at various frequencies (*Figure 4a*). CV in dystrophin-deficient iPSC-CM monolayers was 50% slower than control monolayers paced at 1 Hz (27 ± 2 and 29 ± 4 cm/s in hemizygous Males 1 and 2 cells, respectively, versus 56 ± 3 cm/s in control cells, *Figure 4b, c*). CV of Control 2 monolayers was 42 ± 5 cm/s (*Figure 4—figure supplement 1*). Remarkably, CV in the heterozygous female monolayers was even slower (18 ± 3 cm/s). In all three groups, the CV restitution curve displayed slightly slower velocities at

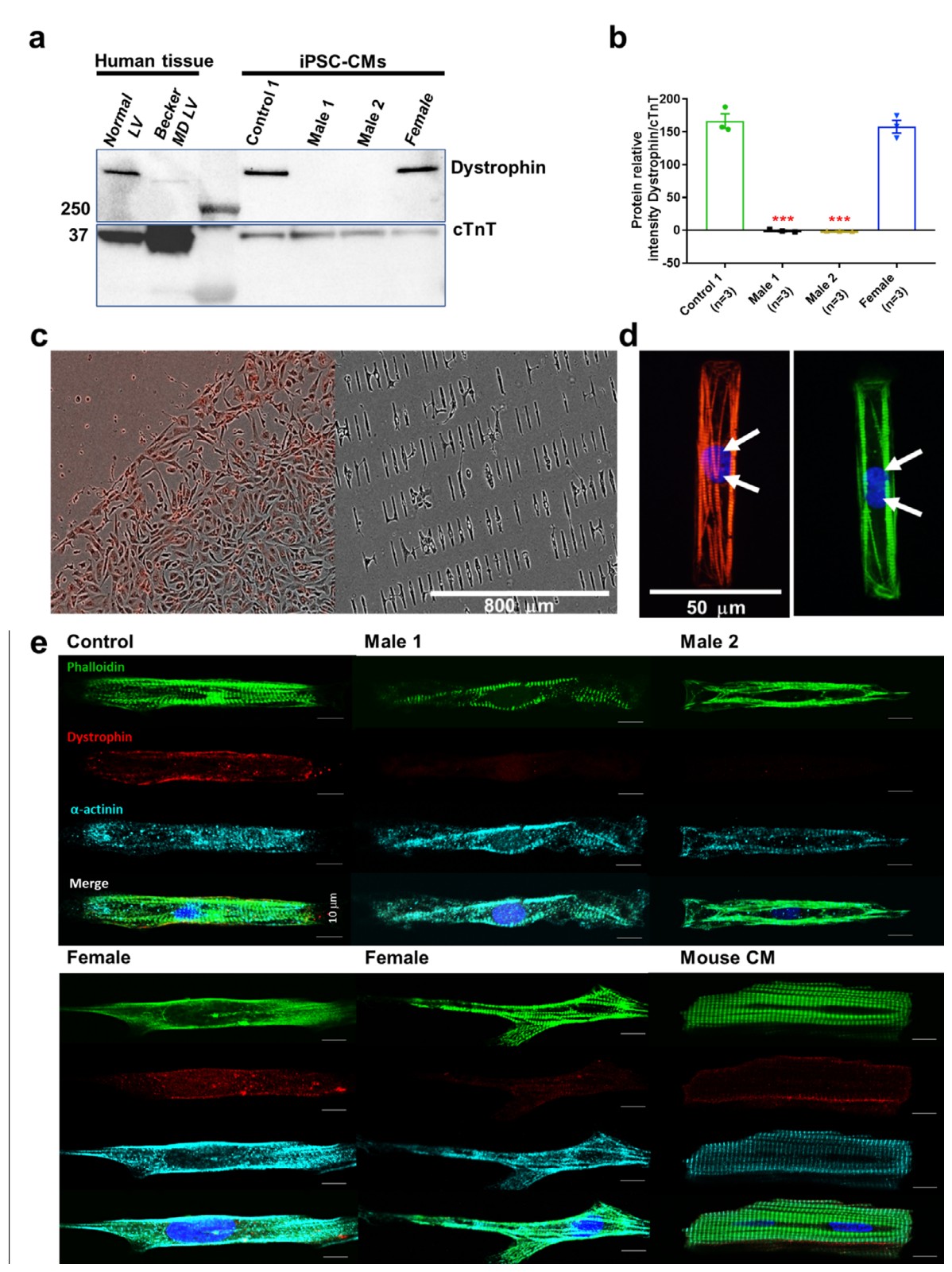

**Figure 2.** Duchenne Muscular Dystrophy (DMD) patient-specific iPSC-CMs do not express dystrophin. (**a**) Top right, control and heterozygous female iPSC-CMs express dystrophin. iPSC-CMs from hemizygous dystrophic cell lines (Males 1 and 2) did not express the large dystrophin isoform. Top left, control tissue lysates from a normal individual and a patient with Becker MD. Dystrophic left ventricular tissue did express dystrophin, but to a lesser extent than normal left ventricle tissue. These tissues were generously provided by the Hypertrophic Cardiomyopathy Clinic, University of Michigan.

*Figure 2 continued on next page*

*Figure 2 continued*

(**b**) Quantitation of dystrophin in control and heterozygous female iPSC-CMs. Dystrophin was absent in DMD iPSC-CMs (p = 0.0001) compared to control iPSC-CMs. Heterozygous female cells exhibited nearly normal dystrophin expression (p = 0.5864). Protein concentration confirmed by western blot against troponin T. Two-tailed Mann–Whitney test. Errors bars, standard error of the mean (SEM). The *n* values are in parentheses. (**c–e**) iPSC-CMs plated onto Matrigel-coated micropatterned PDMS. (**c**) Male 1 iPSC-CMs plated as a monolayer on a Matrigel-coated PDMS (left) for 1 week, and then dissociated for replating onto micropatterned PDMS (right). (**d**) Control iPSC-CMs fixed and stained on micropatterns. Immunostaining for cardiac troponin I (red) and F-actin (green). Nuclei were stained with DAPI (white arrows). Scale bar, 50 µm. (**e**) Immunostaining for dystrophin in iPSC-CMs from control, dystrophic Males 1 and 2, female, and mature mouse cardiomyocytes. DMD cells did not express dystrophin compared to control. Heterozygous female iPSC-CMs showed variable expression of dystrophin. Scale bar, 10 µm.

The online version of this article includes the following source data for figure 2:

**Source data 1.** Full unedited gel.

higher frequencies (*Figure 4d*). Most important, in the female monolayer (*Figure 4e*), slower and more heterogeneous patterns of electrical wave propagation were accompanied by focal discharges in the form of trigeminy (*Figure 4e*, *left*), which often triggered unidirectional block and reentry (*Figure 4e*, *right*). Altogether, the data presented in *Figures 3 and 4* provide a direct mechanistic explanation for the conduction abnormalities and arrhythmias seen on the ECGs of at least two of the patients (see *Figure 1*). In all three iPSC-CMs from affected individuals, the reduced CV occurred in the absence of measurable changes in connexin43 (Cx43) protein (*Figure 4—figure supplement 2*, *Figure 4—figure supplement 2—source data 1*). We did not detect any significant differences in Cx43 expression among control, heterozygous, and hemizygous iPSC-CMs in these monolayer experiments.

## Sodium current is downregulated in DMD iPSC-CMs

Sodium channels determine the upstroke velocity of the cardiac action potential and consequently play a key role in the conduction of the cardiac electrical impulse (*Abriel, 2007*). Here, we compared the sodium current ($I_{Na}$) density in the DMD male and female iPSC-CMs versus each of the controls. In *Figure 5a,b*, *Figure 5—source data 1*, the peak inward $I_{Na}$ density in hemizygous iPSC-CMs was significantly decreased (−14 ± 1 pA/pF for Male 1 cells and −15 ± 1 pA/pF for Male 2 cells) compared to both Control 1 (−27 ± 3 pA/pF) and Control 2 iPSC-CMs (−38 ± 1 pA/pF; *Figure 5—figure supplement 1g*, *Supplementary files 4 and 5*). Importantly, the $I_{Na}$ density in heterozygous female cells was also dramatically reduced (−11 ± 1 pA/pF). Altogether, except for peak sodium current density, statistical comparisons in terms of biophysical properties of $I_{Na}$ (half maximal activation, slope factor, reversal potential) for DMD versus Control 1 (*Supplementary file 4*), DMD versus Control 2 (*Supplementary file 5*), and Control 1 versus Control 2 (*Supplementary file 6*) showed no differences among any of the groups. Also, as shown in *Figure 5—figure supplement 1*, cell capacitance in all the patient-specific cells was similar to control, indicating that cell size was similar in all groups.

The above data indicate that dystrophin deficiency reduces the $I_{Na}$ density, which may be considered one of the main causes for the cardiac conduction defects reported in DMD patients (*Yotsukura et al., 1992*; *Perloff, 1984*). The absence of dystrophin might also affect other ionic currents. For instance, the L-type calcium current ($I_{Ca,L}$) is increased in cardiomyocytes from adult *mdx* mice (*Koenig et al., 2014*; *Viola et al., 2013*). In addition, as previously suggested, $I_{Ca,L}$ density is increased in iPSC-CMs from DMD patients (*Yotsukura et al., 1992*). However, under our experimental conditions, $I_{Ca,L}$ was unaltered in hemizygous and heterozygous DMD iPSC-CMs (*Supplementary files 4 and 6*; *Figure 5—figure supplement 2*). Differences in culture conditions and cell maturation (see Methods, *Herron et al., 2016*) might have contributed to the different outcomes in the two studies.

## DMD iPSC-CMs have reduced inward rectifier potassium currents

Apart from the well-described regulation of Na$_V$1.5 channels by the DAPC (*Gavillet et al., 2006*; *Petitprez et al., 2011*), there is evidence that this protein complex also regulates Kir2.1 inward rectifying potassium channels in *mdx* cardiomyocytes (*Rubi et al., 2017*). Moreover, a pool of Na$_V$1.5 channels colocalizes with Kir2.1 forming protein complexes with scaffolding proteins at the cardiomyocyte lateral membrane and intercalated disc, where they modulate each other's surface expression (*Milstein et al., 2012*; *Matamoros et al., 2016*; *Ponce-Balbuena et al., 2018*). To test whether, in addition to $I_{Na}$, the inward rectifier potassium current is also affected in iPSC-CMs from DMD patients, we compared Ba$^{2+}$-sensitive potassium currents ($I_{K1}$). In *Figure 5c, d*, *Figure 5—source data 1*, $I_{K1}$

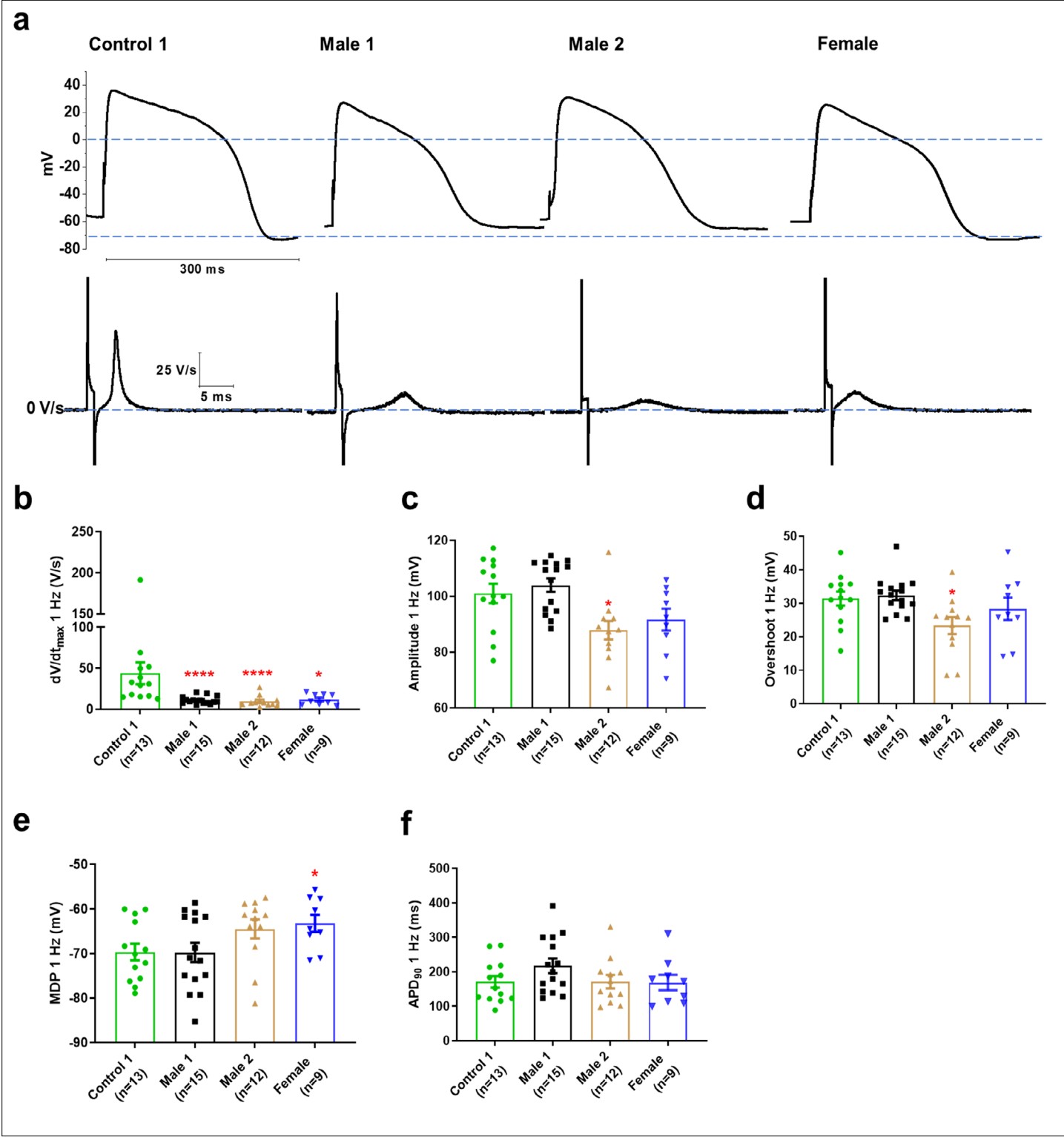

**Figure 3.** Action potential properties in control, Duchenne Muscular Dystrophy (DMD), and female iPSC-CMs. (**a**) Representative action potentials of ventricular-like iPSC-cardiomyocytes from Control 1, heterozygous female, and DMD individuals. The respective d$V$/d$t$ trace is shown below each action potential. (**b**) Mann–Whitney test revealed that d$V$/d$t_{max}$ was reduced in both DMD compared to Control 1. d$V$/d$t_{max}$ was also significantly reduced in the female cells. (**c–e**) Overshoot and amplitude were only affected in the Male 2 iPSC-CMs, while heterozygous female cells were significantly more depolarized compared to control. (**f**) APD$_{90}$ was similar in all iPSC-CMs tested. Cells plated on micropatterns were paced at 1 Hz. Errors bars, standard error of the mean (SEM). The $n$ values are in parentheses. Two-tailed Mann–Whitney test. ****p = 0.0001 and *p <0.05.

*Figure 3 continued on next page*

*Figure 3 continued*

The online version of this article includes the following source data and figure supplement(s) for figure 3:

**Source data 1.** Action potential parameters.

**Figure supplement 1.** Electrophysiological analysis in Control 2 (human foreskin-derived BJ iPSC-CMs).

**Figure supplement 2.** Action potential properties of control, hemizygous, and heterozygous Duchenne Muscular Dystrophy (DMD) iPSC-CMs paced at 2 Hz.

density measured at −120 mV was significantly reduced in Male 1 (−1 ± 0.3 pA/pF) and Male 2 (−1.2 ± 0.3 pA/pF) iPSC-CMs compared to Control 1 (−3.2 ± 0.5 pA/pF). $I_{K1}$ density of Control 2 cells was −2.6 ± 0.6 pA/pF. Changes in $I_{K1}$ were highly variable in heterozygous cells, and the difference with control was not significant, likely due to the variability of expression of dystrophin (*Figure 2e*) and other proteins forming the complex.

## Ion channel gene expression profile in male and female DMD iPSC-CMs

Previous reports have shown that when one of the DAPC components is genetically absent, other proteins of the complex are likewise downregulated, leading to a dysfunction of the complex (*Araishi et al., 1999*). To confirm whether this phenomenon occurs in both hemizygous and heterozygous DMD iPSC-CMs, we analyzed the mRNA levels, and protein expression of the cardiac ion channels $Na_V1.5$ (encoded by *SCN5A* gene), Kir2.1 (encoded by *KCNJ2* gene), and $Ca_V1.2$ (encoded by *CACNA1C* gene).

Consistent with what has been described for mdx mice (*Gavillet et al., 2006*), both hemizygous DMD iPSC-CMs showed increased *SCN5A* expression (*Figure 5—figure supplement 3a*, *top*), also like human cardiac tissue from a Becker MD (BMD) individual (*Figure 5—figure supplement 3a*, *bottom*). Similarly, *KCNJ2* gene expression was upregulated in both hemizygous DMD cell lines, as well as the BMD individual (*Figure 5—figure supplement 3b*). This suggests that the increase in cardiac *SCN5A* and *KCNJ2* mRNA levels might be a general compensatory phenomenon in DMD patients. On the other hand, consistent with the unaffected $I_{CaL}$, neither *CACNA1C* nor $Ca_V1.2$ were modified in either male or female DMD iPSC-CMs compared to control (*Figure 5—figure supplement 3c*).

To test whether the decreased $I_{K1}$ and $I_{Na}$ in both DMD iPSC-CMs were due to reduced $Na_V1.5$ and Kir2.1 protein levels, we performed western blot experiments with total protein lysates of iPSC-CMs monolayers. In *Figure 5—figure supplement 4a,b*, *Figure 5—figure supplement 4—source data 1*, the absence of dystrophin coincided with a consistent reduction of total $Na_V1.5$ protein. Surprisingly, we did not observe any change in total Kir2.1 protein. To investigate whether the reduced $I_{K1}$ and $I_{Na}$ in DMD iPSC-CMs was due to reduced membrane protein levels, we conducted protein biotinylation assays (*Figure 5—figure supplement 4c,d*, *Figure 5—figure supplement 4—source data 1*). Biotinylated $Na_V1.5$ was significantly lower than control in the Male 2 cell line only. Biotinylated Kir2.1 was significantly reduced in the hemizygous cells, consistent with the reduction in $I_{K1}$. Altogether, the results presented thus far support the idea that the absence of dystrophin in the DMD iPSC-CMs resulted in reduced abundance of $Na_V1.5$ protein in the whole cell and likely reduced trafficking of both $Na_V1.5$ and Kir2.1 to the cell membrane, as predicted from our previous work (*Matamoros et al., 2016*; *Ponce-Balbuena et al., 2018*; *Pérez-Hernández et al., 2018*).

The data in iPSC-CMs from the heterozygous female are more challenging. $Na_V1.5$ total protein levels and biotinylated $Na_V1.5$ channels were not different from control (*Figure 5—figure supplement 4*), but the $I_{Na}$ density in single iPSC-CMs was even smaller than in DMD iPSC-CMs. This, together with the lack of significance in the changes of $I_{K1}$ density, total Kir2.1 protein level, and biotinylated Kir2.1, lead us to conclude that the large variability in the expression of dystrophin significantly influenced the overall results in the heterozygous cells.

## α1-Syntrophin expression restores electrophysiological defects in DMD iPSC-CMs

In the heart, the dystrophin-associated protein α1-syntrophin (*SNTA1*) acts as a scaffold for numerous signaling and ion channel proteins that control cardiac excitability (*Finsterer and Stöllberger, 2003*; *Araishi et al., 1999*; *Gee et al., 1998*). α1-Syntrophin is a PDZ domain protein that colocalizes and forms a macromolecular complex ('channelosome') with Kir2.1 and $Na_V1.5$ at the sarcolemma

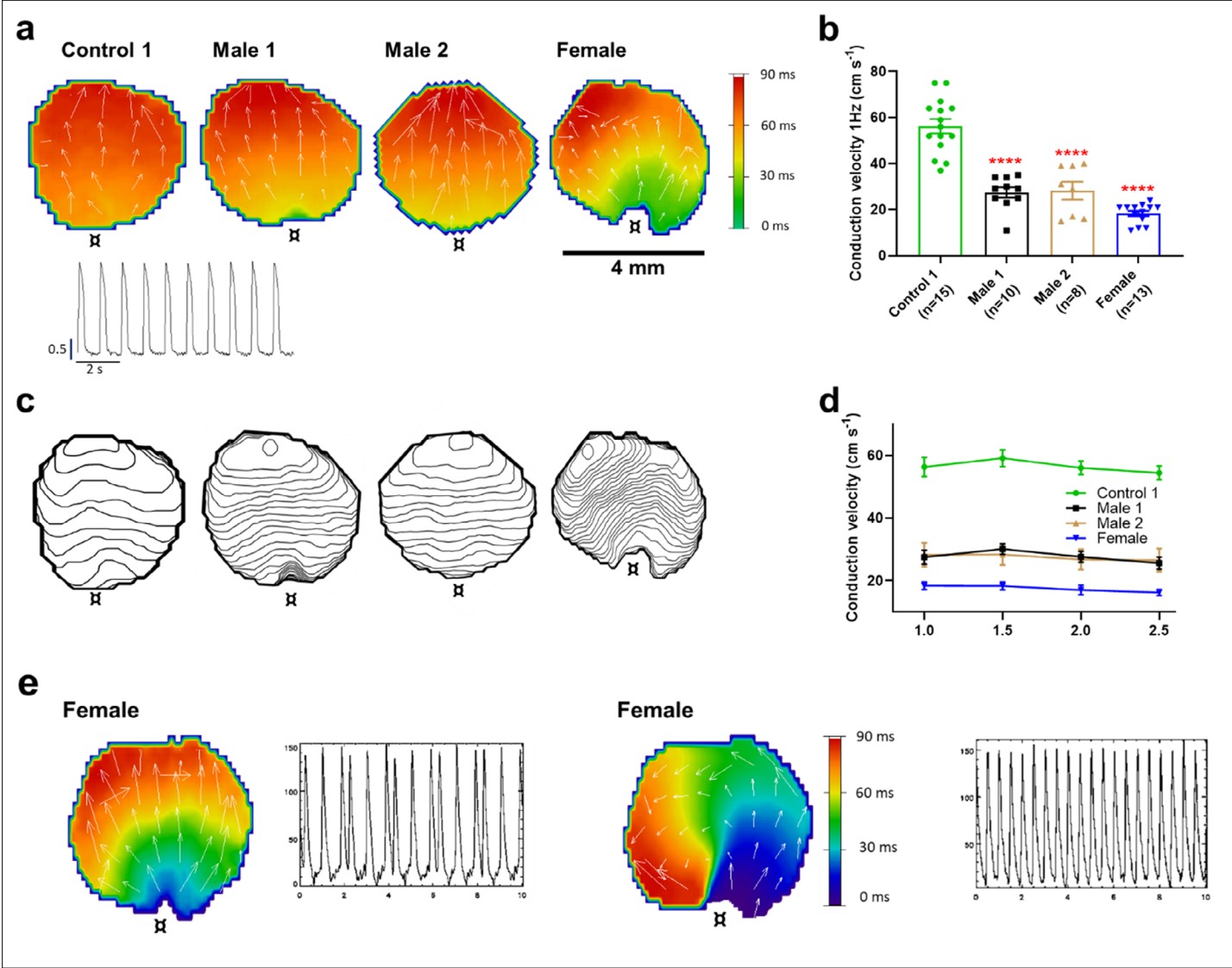

**Figure 4.** Conduction velocity (CV) is slower in iPSC-CM monolayers from Duchenne Muscular Dystrophy (DMD) hemizygous male and heterozygous female than control. (**a**) Activation maps of action potential propagation at 1 Hz. Each color represents a different activation time with time zero appearing in green (¤ indicates the location of the stimuli for each monolayer). White vectors (↑) are a measure of local velocity and direction of the wave. *Inset*. Representative optical action potentials (APs) at 1 Hz. (**b**) Bar graphs of CV in each monolayer group, as indicated. Numbers in parenthesis are number of monolayers per group. (**c**) Averaged 2-ms contour isochrone maps for each representative monolayer above. Tighter averaged isochrone contours in the hemizygous and heterozygous iPSC-CM monolayers indicate slowed and more heterogeneous CV compared to control. (**d**) CV restitution tended to slow in all groups as pacing frequency increased. (**e**) Arrhythmias in heterozygous female iPSC-CMs monolayers. Left map, spontaneous pacemaker activity; *Left inset*, single pixel recording reveals premature ectopic discharges in a pattern of trigeminy; right map, high-frequency reentrant tachycardia maintained by a self-sustaining rotor; *right inset*, single pixel recording shows the interbeat interval (500 ms) of the reentrant tachycardia. Errors bars represent standard error of the mean (SEM). The *n* values are in parentheses. Two-tailed Mann–Whitney test. ****$p < 0.0001$.

The online version of this article includes the following source data and figure supplement(s) for figure 4:

**Figure supplement 1.** Conduction velocity in control BJ iPSC-CM (Control 2) monolayer.

**Figure supplement 2.** Cx43 expression level in control, hemizygous, and heterozygous Duchenne Muscular Dystrophy (DMD) iPSC-CMs.

**Figure supplement 2—source data 1.** Full unedited gel showing Dystrophin, Actinin and Cx43 expression levels by WB technique.

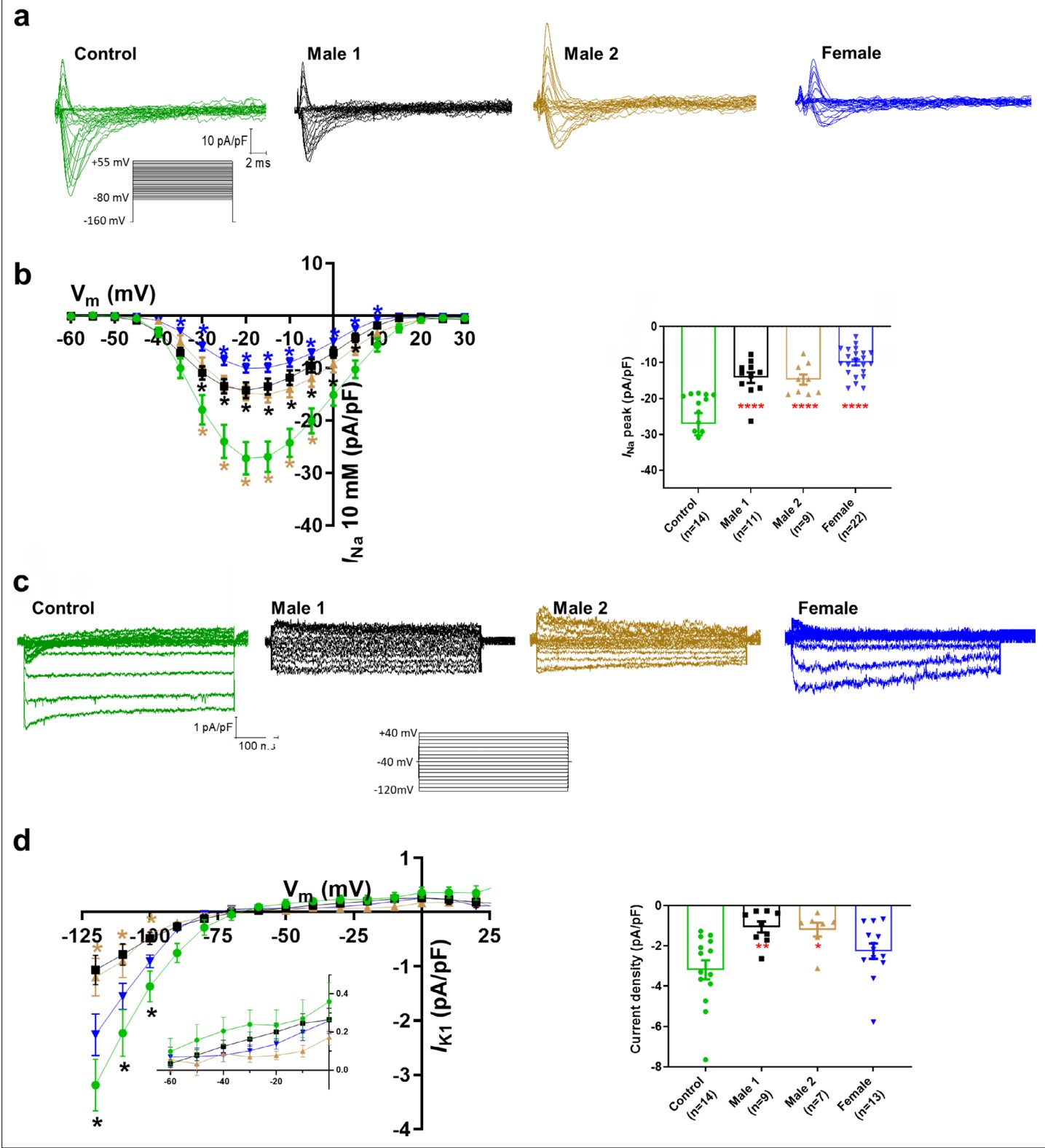

**Figure 5.** Sodium ($I_{Na}$) and Inward rectifier potassium ($I_{K1}$) channel properties in control, Duchenne Muscular Dystrophy (DMD), and female iPSC-cardiomyocytes. (**a**) Superimposed $I_{Na}$ current traces for Control 1, hemizygous, and heterozygous iPSC-CMs elicited by the pulse protocol shown by the inset. (**b**) *Left*, normalized current–voltage (*I/V*) relationships. $I_{Na}$ was significantly reduced in both Males 1 and 2 iPSC-CMs compared with control at the specified voltages. Heterozygous female iPSC-CMs showed also a very reduced current density from 35 to 10 mV. Two-way analysis of variance (ANOVA) followed by Sidak's multiple comparisons test. *Right,* peak $I_{Na}$ density at 20 mV was reduced in all three affected groups compared to control. (**c**) Typical

*Figure 5 continued on next page*

*Figure 5 continued*

$I_{K1}$ density traces from control and DMD cells elicited by the pulse protocol in the *inset*. (**d**) *Left, I/V* relationships. $I_{K1}$ was significantly reduced in both Males 1 and 2 iPSC-CMs compared with control at the specified voltages. Two-way ANOVA followed by Sidak's multiple comparisons. *Right*, normalized current densities at −120 mV. $I_{K1}$ was decreased in Males 1 and 2 cells compared to control cells. Two-tailed Mann–Whitney test. Errors bars represent standard error of the mean (SEM). The *n* values are in parentheses. ****$p < 0.0001$, **$p < 0.005$, and *$p < 0.05$ and *$p < 0.056$.

The online version of this article includes the following source data and figure supplement(s) for figure 5:

**Source data 1.** Voltage clamp data.

**Figure supplement 1.** Cell capacitance.

**Figure supplement 2.** Calcium channel properties in control, Duchenne Muscular Dystrophy (DMD), and female iPSC-CMs.

**Figure supplement 3.** *SCN5A, KCNJ2,* and *CACNA1C* mRNA expression in control, Duchenne Muscular Dystrophy (DMD), and female iPSC-CMs.

**Figure supplement 4.** $Na_V1.5$ protein level is significantly reduced in patient-specific Duchenne Muscular Dystrophy (DMD) iPSC-CMs.

**Figure supplement 4—source data 1.** Full unedited gel.

(*Petitprez et al., 2011*; *Matamoros et al., 2016*; *Gee et al., 1998*; *Milstein et al., 2012*). Since α1-syntrophin has been shown to modify $I_{Na}$ and $I_{K1}$ by enhancing membrane $Na_V1.5$ and Kir2.1 membrane levels (*Matamoros et al., 2016*), we hypothesized that even in the absence of dystrophin, increasing α1-syntrophin should restore normal electrical function in the DMD iPSC-CMs. Therefore, we stably transfected *SNTA1* gene via piggyBac transposon-based mammalian cell expression system in Male 1 cells verifying an increase in syntrophin expression (*Figure 6a, b*). As illustrated in *Figure 6—source data 1*, α1-syntrophin expression increased the Kir2.1 and $Na_V1.5$ protein levels in the membrane fraction as indicated by colocalization with wheat germ agglutinin (WGA) compared to controls transfected with GFP. In *Figure 7—source data 1*, α1-syntrophin expression resulted in a recovery of both $I_{Na}$ (*Figure 7a*) and $I_{K1}$ (*Figure 7b*). Consequently, as shown in *Figure 7—source data 1*, *SNTA1* transfection led to significant improvement in the electrophysiological properties of DMD iPSC-CMs. The MDP was hyperpolarized, the $dV/dt_{max}$ and amplitude were increased and the $APD_{90}$ was abbreviated.

## α1-Syntrophin expression prevents reentrant arrhythmias in DMD iPSC-CMs

We conducted additional optical mapping experiments in iPSC-CM monolayers from control 1, Male 1, and Male 1 + SNTA1. First, we confirmed that Male 1 monolayers had a significantly reduced CV. However, while *SNTA1* expression had a variable effect from monolayer to monolayer, by helping to increase $Na_V1.5$ and Kir2.1 at the cell membrane, it had a strong tendency to increase CV toward control (*Figure 8a,b*, *Figure 8—source data 1*). In addition SNTA1 expression significantly shortened the prolonged Male 1 optical APD, returning it to control (*Figure 8c,d*, *Figure 8—source data 1*). Most important, as demonstrated by the data presented in *Figure 8e*, as well as *Video 1*, lack of dystrophin in Male 1 iPSC-CM monolayers significantly increased the inducibility of reentrant arrhythmias (rotors), yielding very complex and highly variable patterns of nonlinear wave propagation throughout the monolayer. These data recapitulate episodes of ventricular flutter and ventricular fibrillation recorded by ECG in the patient (*Figure 1b*). As summarized in *Figure 8f*, *SNTA1* transfection eliminated arrhythmia inducibility in 9/10 monolayers from this patient.

## α1-Syntrophin expression increases contractility in DMD iPSC CMs

We have conducted additional experiments to measure contractility (i.e., cell shortening) in single iPSC-CMs a well as iPSC-CM monolayers, as illustrated in *Figure 8—figure supplement 1*. We demonstrate that Male 1 iPSC-CMs have a significant defect in contractility, manifested as prolonged shortening and relaxation times, reduced shortening slope and reduced shortening amplitude. Importantly, *SNTA1* expression rescues all contractility parameters (*Figure 8—figure supplement 1b–e*). These data highlight the importance of the $Na_V1.5$–Kir2.1 channelosome function in helping to ensure a rapid action potential upstroke velocity leading to an adequate excitation–contraction coupling and contraction.

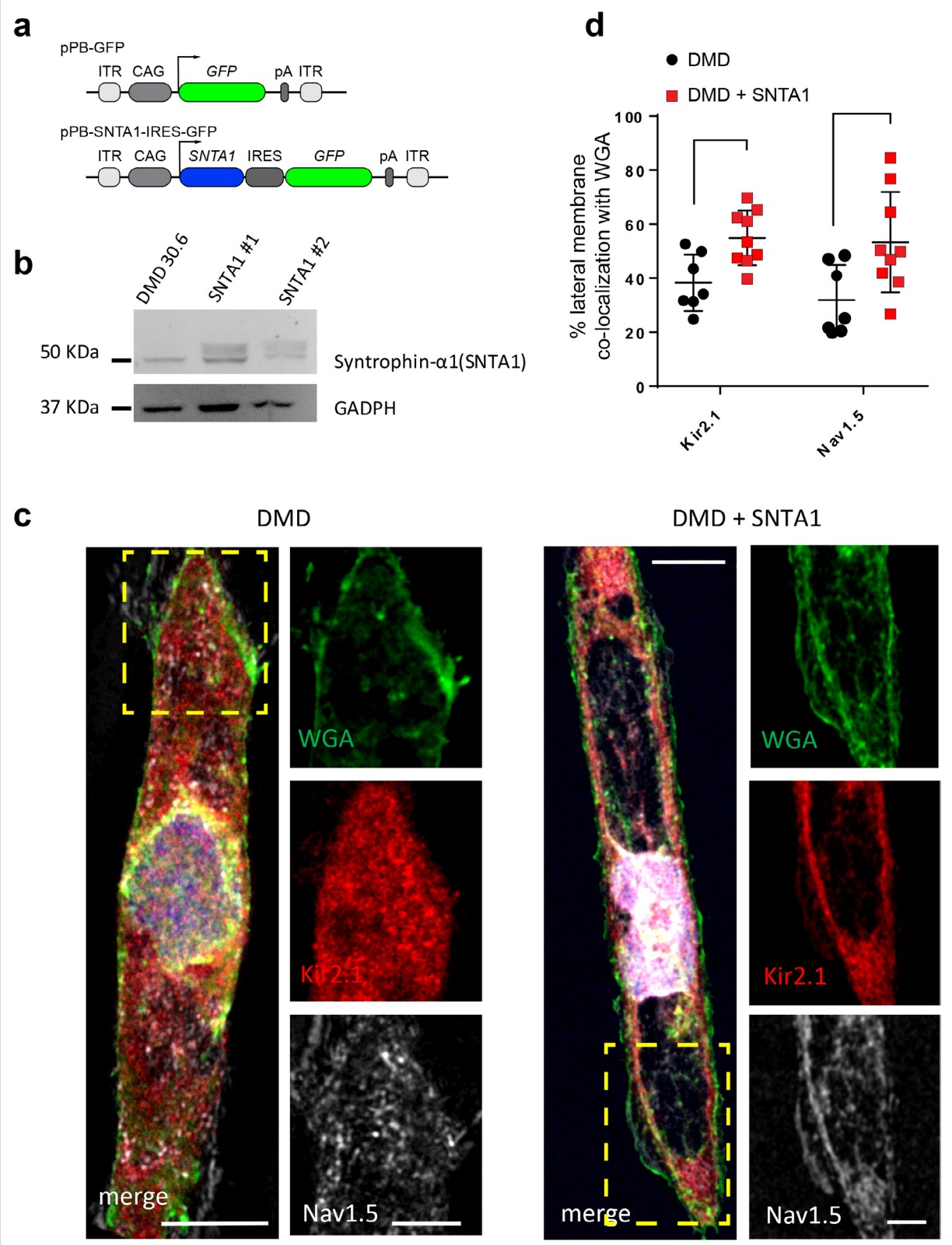

**Figure 6.** Transfection of *SNTA1* rescues membrane levels of Kir2.1 and Na$_V$1.5 proteins in iPSC-CMs from Male 1 patient. (**a**) Cartoon illustrating nonviral piggy-bac vector encoding *SNTA1* for transfection in Male 1 iPSC-CMs. *SNTA1* coding region (CDS) is driven by the CAG promoter and followed by green fluorescence protein (GFP) after an internal ribosome entry site (IRES). Control vector only expresses GFP. (**b**) Western blot for α1-syntrophin expression normalized with GAPDH. (**c**) Immunostaining for Kir2.1 (red), Na$_V$1.5 (white), and wheat germ agglutinin (WGA; green) in control

*Figure 6 continued on next page*

*Figure 6 continued*

Male 1 iPSC-CM (left) and Male 1 iPSC-CM transfected with *SNTA1*. Nuclei were stained with DAPI. Yellow arrows point to iPSC-CM membrane staining. Scale bar, 5 µm. (**d**) Quantification of Kir2.1 and Na$_V$1.5 colocalization with WGA at the cell membrane shows significant increase of both Kir2.1 (*$p < 0.05$; $n$ = 7–10 cells) and Na$_V$1.5 (**$p < 0.01$; $n$ = 7–10 cells).

The online version of this article includes the following source data for figure 6:

**Source data 1.** SNTA1 quantification (*Figure 6d*).

## Discussion

We demonstrate here that patient-specific iPSC-CMs recapitulated consistently the hallmark electrophysiologic features of cardiomyopathic DMD patients (*Finsterer and Stöllberger, 2003*), In fact, mature iPSC-CMs from two hemizygous male DMD patients lacking the Dp427 isoform and a female patient heterozygous for a 5-exon deletion (Δ8–12) in the dystrophin gene have significantly reduced $I_{Na}$ and $I_{K1}$ densities, d$V$/d$t_{max}$ and conduction velocities, as well as focal and reentrant arrhythmias. Together, these results strongly suggest that reduced excitability underlies the arrhythmogenic mechanism in DMD patients. While all patients developed severe cardiomyopathy, they also suffered frequent PVCs and ventricular tachycardia. In addition, the ECG of the heterozygous female DMD patient showed a significant left axis deviation caused by cardiac conduction defects in line with our results. In one of the male patients, ICD recordings revealed the arrhythmia deteriorating into ventricular fibrillation (*Hara et al., 2016*). Our results in patient-specific iPSC-CMs indicate that such defects are a direct consequence of a Na$_V$1.5–α1-syntrophin–Kir2.1 channelosome dysfunction produced by the disruption of the DAPC that characterizes the DMD cardiomyopathy. Remarkably, transfecting just one of the components of that complex (i.e., α1-syntrophin) in Male 1 iPSC-CMs led to channelosome recovery at the plasma membrane, with restoration of $I_{Na}$ and $I_{K1}$ densities, MDP, AP d$V$/d$t_{max}$, and amplitude. Moreover, in iPSC-CM monolayers from the same patient, α1-syntrophin transfection tended to restore rapid conduction, reduced APD duration and prevented reentrant arrhythmia inducibility. To our knowledge, this report is first in providing a comprehensive and rigorous mechanistic demonstration of the potential causes of cardiac conduction defects and arrhythmogenesis in human DMD, substantially extending findings from animal models (*Gavillet et al., 2006*).

ECG abnormalities can be detected in up to 60% of DMD patients (*Finsterer and Stöllberger, 2003*), and among those, conduction defects, bradycardia, ventricular arrhythmias, and sudden death are frequent (*Perloff, 1984*). However, despite significant progress in the understanding of the mechanisms of the skeletal muscle dystrophy, exploration of the electrophysiological consequences of the dystrophic cardiomyopathy has been slower. Until now, it has been difficult to link functional changes in individual ion channels/proteins with corresponding clinical phenotypes in inheritable ion channel diseases and cardiomyopathies such as DMD (*Villa et al., 2015*).

Both Na$_V$1.5 and Kir2.1 interact with the DAPC via α1-syntrophin through their respective canonical C-terminal PDZ-binding domains. As shown previously, Na$_V$1.5 has an additional internal PDZ-like binding domain localized at the N-terminus that also interacts with α1-syntrophin (*Gavillet et al., 2006*; *Matamoros et al., 2016*). Changes in $I_{Na}$ and $I_{K1}$ might alter cardiac conduction and increase the probability of premature beats like those seen on the ECG from the DMD patient (*Gavillet et al., 2006*). We showed here that in addition to reduced $I_{Na}$, iPSC-CMs from DMD patients also have reduced $I_{K1}$ and probably alterations in other proteins altogether causing proarrhythmic alteration in electrical impulse conduction, likely because of trafficking disruption of the α1-syntrophin-mediated macromolecular complex formed by the DACP with Kir2.1and Na$_V$1.5. The important role of such an ion channel complex in controlling cardiac electrical function is highlighted by our demonstration that lack of the DAPC in the iPSC-CMs from the DMD patient reduces CV and that expression of α1-syntrophin rescues excitability, and normal action potential characteristics, thus preventing reentrant arrhythmias. In addition, the demonstration that α1-syntrophin also restores contractility in these patient-specific cells highlights the importance of the Na$_V$1.5–α1-syntrophin–Kir2.1 channelosome in ensuring a rapid and well-coordinated coordinated coupling between excitation and contraction. Altogether, our results provide a straightforward arrhythmogenic mechanism in DMD-associated cardiomyopathy and offers a potential treatment.

Maturation of iPSC-CMs is essential for human disease modeling and preclinical drug studies (*da Rocha et al., 2017*; *Ronaldson-Bouchard et al., 2019*). Culturing iPSC-CM monolayers on soft

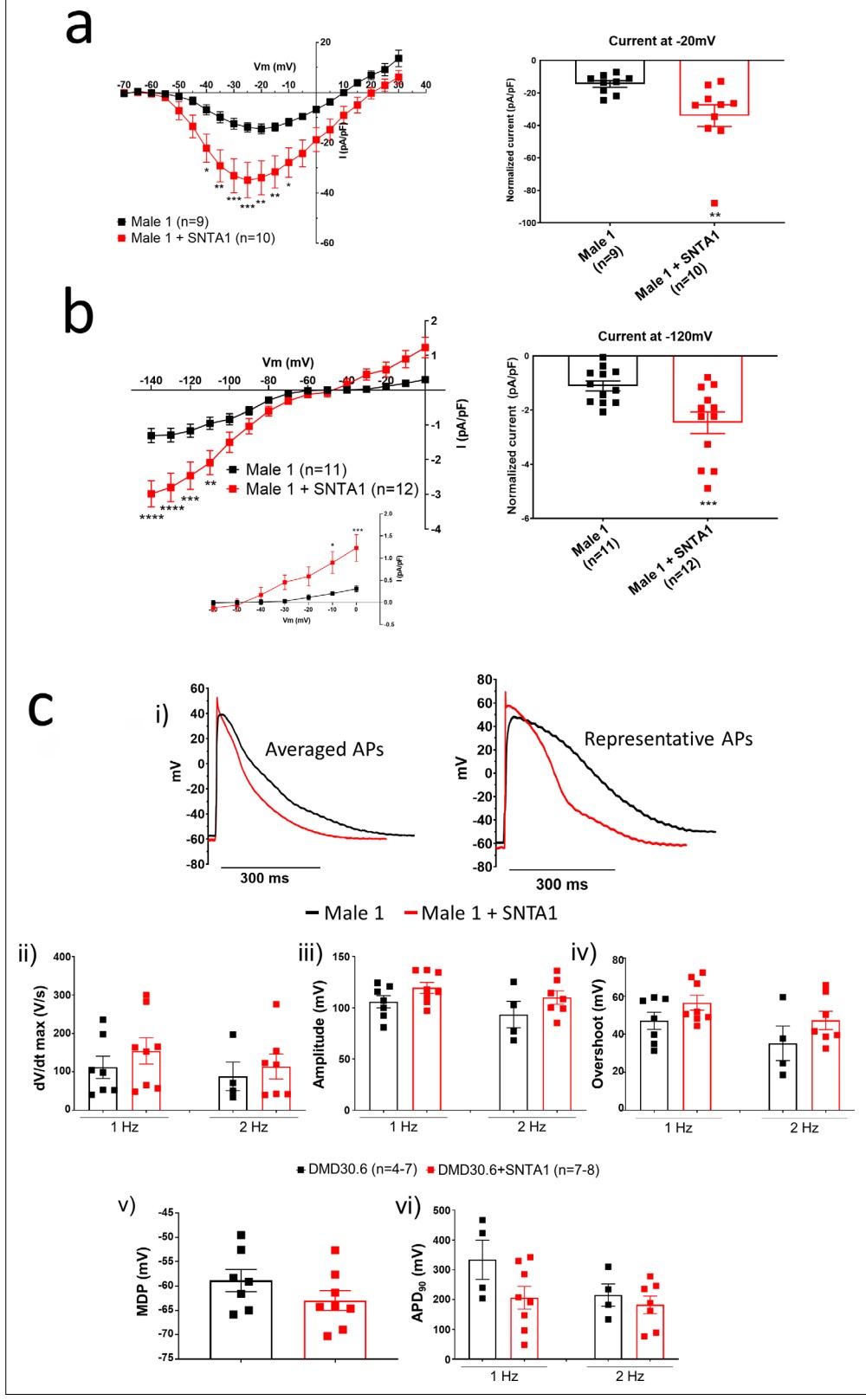

**Figure 7.** *SNTA1* expression restores the electrophysiological deficiencies in Duchenne Muscular Dystrophy (DMD) iPSC-CMs. (**a**, **b**) Normalized current–voltage (*I/V*) relationships for $I_{Na}$ and $I_{K1}$ in Male 1 before (black) and after (red) syntrophin expression at the specified voltages. Two-way analysis of variance (ANOVA) followed by Sidak's multiple comparisons test. Graphs show peak $I_{Na}$ density at −20 mV (**a**) and peak $I_{K1}$ density at −120 mV (**b**). The inset in B

*Figure 7 continued on next page*

*Figure 7 continued*

highlights the increased outward component of $I_{K1}$ at less negative potentials upon syntrophin expression. Two-tailed Mann–Whitney test. (**c**) Effect of syntrophin expression on AP showing: (**i**) Averaged (*left*) and representative (*right*) action potential traces of ventricular-like iPSC-cardiomyocytes derived from DMD cells before (black) and after (red) syntrophin expression, (**ii**) maximal AP upstroke velocity ($dV/dt_{max}$), (**iii**) amplitude, (**iv**) overshoot, (**v**) MDP, and (**vi**) $APD_{90}$. Errors bars represent standard error of the mean (SEM). The *n* values are in parentheses. *p < 0.05; **p < 0.01; ***p < 0.001; and p < 0.0001.

The online version of this article includes the following source data for figure 7:

**Source data 1.** SNTA1 IV IK1 and IV INA (*Figure 7A and B*); and, Action potential properties (*Figure 7C*).

PDMS membranes coated with Matrigel promotes cell maturation (*Herron et al., 2016*). Also, there are several reports indicating that the regulation of cell shape and substrate stiffness helps improve the contractile activity and maturation of iPSC-CMs (*Kuo et al., 2012*; *Ribeiro et al., 2015*). Thus, having cells with ventricular-like action potentials and structural and electrophysiological maturity that approximates the human adult ventricular cardiomyocyte is likely to be more useful in investigating the pathophysiology of DMD patients. Therefore, here we used a micropatterning platform based on Matrigel-coated PDMS membrane (*Herron et al., 2016*) for modeling single-cell cardiac electrical activity. Our findings showed that culturing single ventricular-like iPSC-CMs on micropatterned Matrigel-coated PDMS confers a cylindrical shape yielding iPSC-CMs with structural and functional phenotypes close to those in human mature cardiomyocytes (*Taggart et al., 1996*; *Grandi et al., 2011*). Electrophysiological analyses in this scenario revealed abnormal action potential profiles in DMD iPSC-CMs, compatible with the clinical alterations observed in both Male 1 and female DMD patients. The strong reduction in $I_{Na}$ density yielded a significant slowing of $dV/dt_{max}$, considered to be an indirect measure of the available functional sodium channels (*Berecki et al., 2010*). Reduction in $I_{Na}$ density was consistent with the relative loss of total $Na_V1.5$ protein levels, and helped us explain the reduced CV in iPSC-CMs from DMD patients. Like other studies (*Petitprez et al., 2011*; *Sanford et al., 2005*), we did not find any change in Cx43 protein levels.

QRS widening and QTc prolongation displayed on the ECGs from the DMD patients are likely related to the changes in functional expression of $Na_V1.5$ and Kir2.1 we have observed in their iPSC-CMs. Both QRS widening and QT dispersion are risk factors for arrhythmias in patients with DMD, and have been implicated in the genesis of ventricular arrhythmias (*Okin et al., 2000*). Interestingly some of the AP parameters of the hemizygous Male 2 iPSC-CMs, including $dV/dt_{max}$, AP amplitude, and overshoot (*Supplementary file 1*), were substantially more reduced than Male 1 and the heterozygous female iPSC-CMs. Such differences are possibly due to the specific mutation in the dystrophin gene. Thus, depending on the mutation in the dystrophin gene each male or female DMD patient might develop different types or levels of cardiac electrical dysfunction and life-threatening arrhythmias.

$I_{Na}$ reduction coincided with $I_{K1}$ reduction in both hemizygous DMD iPSC-CMs, supporting the idea that both channels require PDZ-mediated interaction with components of the DAPC to modulate reciprocally their proper expression (*Gavillet et al., 2006*; *Leonoudakis et al., 2004*). It is likely that the reduced $I_{K1}$ in the DMD iPSC-CMs contributed to the reduced $dV/dt_{max}$, although the MDP in the iPSC-CMs from the two dystrophic patients was like control. In this regard, it is important to note that the relationship between MDP and $I_{Na}$ availability is highly nonlinear in such a way that a very small reduction in MDP is expected to result in substantial reduction in sodium current during the action potential upstroke (*Shaw and Rudy, 1997*). Regardless, the biotinylation experiments demonstrated that Kir2.1 levels at the membrane were significantly lower in both DMD iPSC-CMs with respect to the control. The elevated *SCN5A* and *KCNJ2* mRNA levels excluded the possibility that a decrease in gene expression was responsible for the protein loss, and therefore, to smaller $I_{Na}$ and $I_{K1}$ densities in the DMD iPSC-CMs. This somehow contrasts with reports in $mdx^{5cv}$ mouse hearts, where the $Na_V1.5$ mRNA levels remained unchanged with a strong reduction in the $Na_V1.5$ protein levels (*Gavillet et al., 2006*). As such, the reduction in the $Na_V1.5$ and Kir2.1 protein levels could be related to ubiquitylation and proteasome degradation as suggested previously in studies in dystrophin-deficient $mdx^{5cv}$ mice (*Rougier et al., 2013*). However, our results in DMD iPSC-CMs strongly suggest that disruption of the DAPC due to lack of dystrophin significantly impairs ion channel expression and function (*Gavillet et al., 2006*; *Koenig et al., 2011*; *Albesa et al., 2011*). Specifically, we demonstrate that the decrease in ion channel current densities is the result of $Na_V1.5$ and Kir2.1 trafficking and membrane

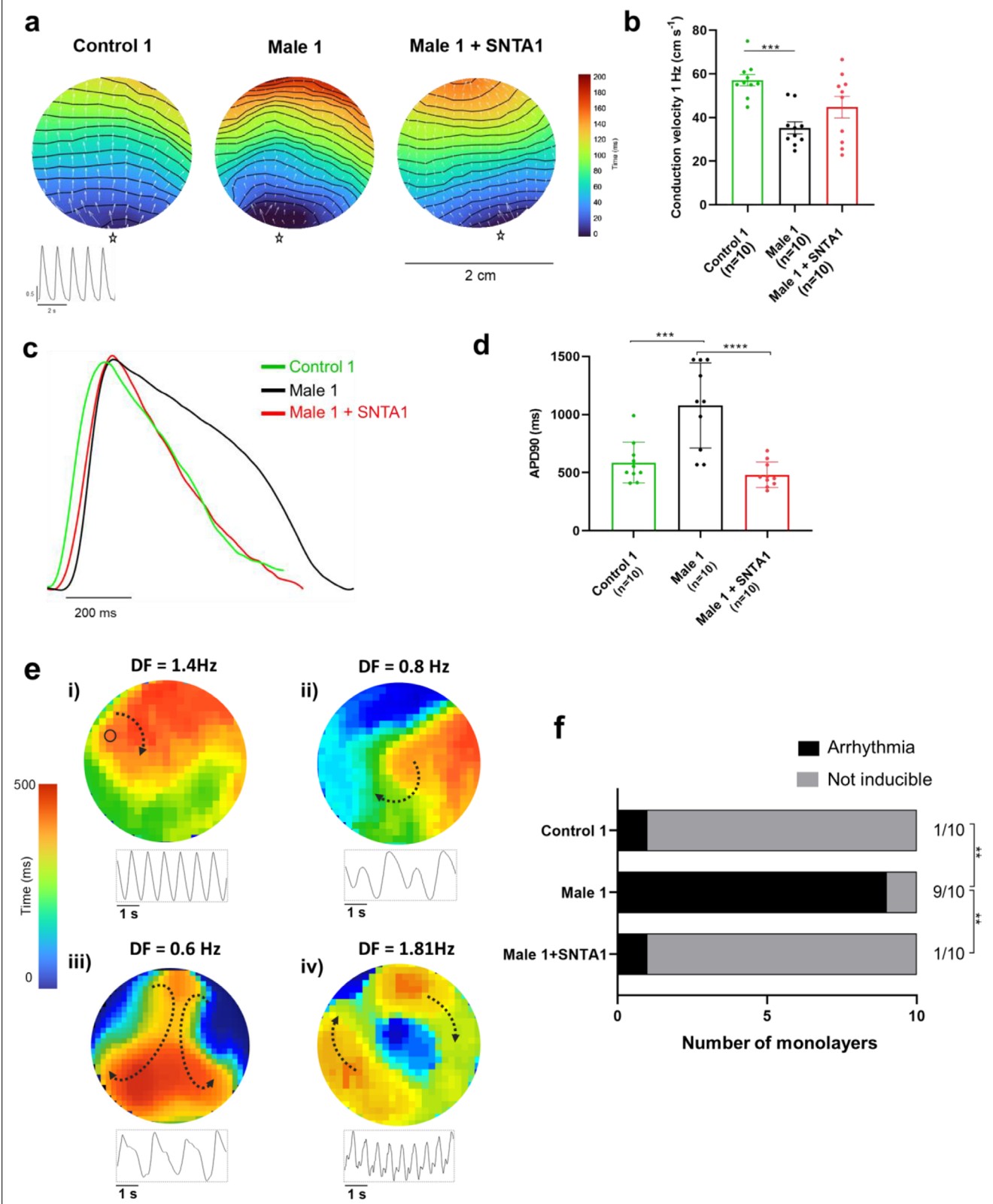

**Figure 8.** SNTA1 expression prevents reentrant arrhythmias in Duchenne Muscular Dystrophy (DMD) iPSC-CM monolayers. (**a**) Optical maps of action potential propagation during pacing at 1 Hz. Each color represents a different activation time with time zero appearing in blue (star below each monolayer indicates the location of the stimuli). Horizontal wavy lines are 10-ms isochrones. White vectors (↑) are a measure of local velocity and direction of the propagation wave. Inset, five representative optical action potentials during 5 s. (**b**) Bar graphs of conduction velocity (CV) in each

*Figure 8 continued on next page*

*Figure 8 continued*

monolayer group, as indicated. Numbers in parenthesis are number of monolayers per group. (**c**) Superimposed mean action potential traces for each representative monolayer. (**d**) Bar graphs of action potential duration at 90% repolarization (APD$_{90}$) for each group of monolayers. (**e**) Reentrant arrhythmias in iPSC-CMs monolayers from DMD hemizygous Male 1 (see also *Video 1i–v*). Below each map is a single pixel recording revealing varying patterns of monomorphic or polymorphic reentrant tachycardia maintained by one (maps i and ii) or more (maps iii and iv) self-sustaining rotors of varying rotation frequency. (**f**) Arrhythmia inducibility for each group shows a high rate of arrhythmia susceptibility in DMD male iPSC-CM monolayers. Error bars represent standard error of the mean (SEM). The *n* values are in parentheses. Two-way analysis of variance (ANOVA) followed by Sidak's multiple comparisons test. Fisher's exact test performed for the arrhythmia inducibility data. ***$p < 0.001$, ****$p < 0.0001$.

The online version of this article includes the following source data and figure supplement(s) for figure 8:

**Source data 1.** Conduction Velocity and APD90 in the absence and in the presence of SNTA1 expression.

**Figure supplement 1.** *SNTA1* expression rescues contractility in Duchenne Muscular Dystrophy (DMD) iPSC-CM monolayers.

targeting defects directly derived from the absence of dystrophin. Such defects can be completely reverted by α1-syntrophin expression, as demonstrated by increases in $I_{Na}$ and $I_{K1}$, and restoration of MDP, action potential upstroke velocity and action potential amplitude, as well as APD abbreviation. On the other hand, the fact that both $I_{Na}$ and $I_{K1}$ are only partially reduced in the DMD iPSC-CMs suggests the presence of different pools of Na$_V$1.5 and Kir2.1 channels that do not depend on DAPC integrity. Altogether, our results support the idea that DMD cardiomyopathy results in ion channel dysfunction that predisposes the dystrophic ventricular myocardium to arrhythmia with potentially lethal consequences.

Previous reports indicate that although heterozygous DMD females have negligible skeletal muscle symptoms, they are not free of cardiac involvement (*Florian et al., 2016*). For example, the clinical expression of the X-linked DMD cardiomyopathy of heterozygous females increases with age (*Florian et al., 2016*). The female patient represented in this study suffered from a relative severe phenotype, characterized by skeletal myopathy and cardiomyopathy, which could be explained by a malignant mutation disrupting the N-terminal of the dystrophin gene. One could assume that one gene of dystrophin should produce enough dystrophin to preserve function in multinucleated skeletal muscle of females (*Holloway et al., 2008*). Unexpectedly, we found that $I_{Na}$ density in iPSC-CMs from the heterozygous female was even more reduced compared to hemizygous iPSC-CMs. Interestingly, the QRS duration was significantly prolonged on the ECG from the heterozygous female compared to the hemizygous patient (see *Figure 1*), suggestive of a more dramatic loss-of-function effect on Na$_V$1.5 in heterozygous females. Probably this is related to the heterogeneity seen in immunostaining studies where some heterozygous female cells express normal dystrophin levels while others show absence or very low expression likely due to random X-inactivation of the WT allele (*Eisen et al., 2019*). Because of random inactivation of one of the X chromosomes, heterozygous females should constitute a mosaic of two or more cell types dramatically differing in the extent of dystrophin expression. Thus, it would not be surprising that females with DMD are more prone to suffer arrhythmias because of spatial electrical inhomogeneity due to variable expression of the mutant allele. The heterogeneity in dystrophin expression has been also observed in canine *carrier* models of X-linked dystrophy, which exhibit a cardiac mosaic pattern, where dystrophin in each myocyte is either fully expressed or absent (*Kane et al., 2013*). Nevertheless, the importance of abnormal cardiac measures in heterozygous females who harbor mutations in the dystrophin gene remains debatable (*Mccaffrey et al., 2017*).

Even though $I_{Na}$ density was substantially reduced in the heterozygous iPSC-CMs, neither the total Na$_V$1.5 protein levels nor the biotinylated Na$_V$1.5 showed any changes. Probably,

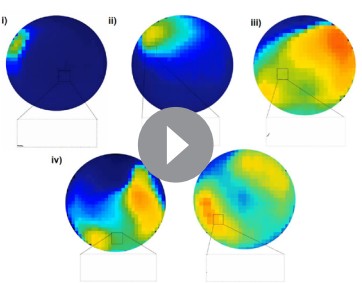

**Video 1.** Duchenne Muscular Dystrophy (DMD) male arrhythmia patterns. Movie # *i* Control iPSC-CM monolayer showing slow and organized spontaneous activity. Movies # *ii–v*. DMD hemizygous Male 1 iPSC-CM monolayers are inducible of high-frequency reentry maintained by one (videos # *ii* and *iii*) or more (videos # *iv* and *v*) self-sustaining rotors mimicking ventricular tachycardia patterns with varying degrees of complexity. Lower tracings in each movie are single pixel recordings from sites indicated by the inset.
https://elifesciences.org/articles/76576/figures#video1

the variable expression of dystrophin in female individuals results in variable Na$_V$1.5 protein levels, while Kir2.1 expression and function are modulated positively to help trafficking of the few pools of Na$_V$1.5 channels belonging to the remaining DAPC. Another possibility that might explain the reduced $I_{Na}$ in heterozygous iPSC-CMs is that the cells may lack a suitable compensatory response due to DAPC disorganization and malfunction. The chimeric nature of the dystrophin mutation in those cells likely makes it more difficult to support a compensatory mechanism than the complete absence of the DAPC complex as it occurs in dystrophic cells. Nonetheless, the very reduced $I_{Na}$ and slowed CV reported in the present study perfectly correlates with the clinical data from the heterozygous female patient. Prolonged QRS duration is evidence of slowed ventricular activation and inhomogeneous conduction and might be associated with rotor activity as observed in both *female* and Male 1 iPSC-CMs monolayers, which is considered a substrate for reentrant ventricular tachycardia (*Richards et al., 1991*). This becomes important because although controversial, heterozygous females may have an age-related increased risk of cardiac conduction disease and sudden death; in female patients of X-linked Emery–Dreifuss muscular dystrophy cardiac alterations typically occur late in life (*Madej-Pilarczyk, 2018*).

## Limitations

We have derived data from experiments conducted in iPSC-CMs from patients who carry independent dystrophin mutations and two unrelated controls, which may be a potential limitation of our study. The original study design included siblings for each DMD cell line. However, getting more experimental groups from the same family was not possible. Nevertheless, both DMD lines lack dystrophin, which gives credence to the idea that loss of dystrophin is important to the shared electrophysiological phenotype independently of the specific mutation. Further, we show new insight into how heterozygous DMD females might show a wide range of cardiac involvement, ranging from asymptomatic to severely impaired electrical cardiac function, particularly the highly reduced $I_{Na}$ leading to slowing of CV, which is reflected on the ECG from the female patient. Thus, together with the structural alterations, the electrophysiological changes may contribute to left ventricular dysfunction in female DMD patients (*Lang et al., 2015*). However, the impact of the finding that the female carrier of the mutation presents a decrease in $I_{Na}$ is somehow mitigated by the fact that since she carries a different mutation, it is difficult to define how the reduction of the $I_{Na}$ in the female carrier compares with the reduction observed in the affected individuals.

iPSC-CMs show significant differences with adult ventricular cardiomyocytes and are still far from recapitulating chamber-specific and layer specific electrical phenotypes of the normal or dystrophic heart. In addition, we cannot generalize our results to patients with different dystrophic gene mutations, such as those underlying Becker muscular dystrophy, which lead to partially truncated dystrophins and may retain specific functional properties of full-length dystrophin. However, enrolling a Becker MD patient was not possible. Also, our syntrophin-mediated rescue experiments were limited to the Male 1 iPSC-CMs line. While caution should be exerted when attempting to extrapolate to the other two DMD cell lines, it is important that the functional defects in the Na$_V$1.5–Kir2.1 channelosome were very similar in the iPSC-CMs from all three patients, which gives credence to our interpretation. Altogether, despite some inherent limitations, our findings provide important mechanistic details on DMD arrhythmogenesis and a crucial lead for investigators interested in developed therapeutic solutions for a deadly disease.

## Acknowledgements

We thank Dr. Adam Helms for help in the implementation of the micropattern technology. We also thank Joseph R Dickens from the CSCAR-University of Michigan for his valuable guidance in the statistical analysis. Dov Freimark MD (from the Leviev Heart Center, Sheba Medical Center, Tel Hashomer, and Tel Aviv University, Israel) supported this study by caring for and enrolling the female patient. We thank Dr. Giovanna Giovinazzo and the staff of the CNIC Pluripotent Cell Unit for their help in processing the iPSCs used in the syntrophin-mediated rescue experiments. We are grateful to patients who despite having a lethal disease agreed to undergo skin biopsy for the sake of science.

## Additional information

### Competing interests

André Monteiro da Rocha: Stembiosys consulting fees and stock options. Todd J Herron, José Jalife: Stembiosys stock options. The other authors declare that no competing interests exist.

### Funding

| Funder | Grant reference number | Author |
|---|---|---|
| National Heart, Lung, and Blood Institute | R01 HL122352 | José Jalife |
| Fundación La Marató 2020 | 736/C/2020 | José Jalife |
| Instituto de Salud Carlos III | PI20/01220 | José Jalife |
| American Heart Association | 19POST34380706s | Eric N Jimenez-Vazquez |
| Israel Science Foundation | 842/19 | Michael Arad |
| Rappaport Foundation | 01012020RI | Ofer Binah |
| Niedersachsen Foundation | ZN3452 | Ofer Binah |
| Horizon 2020 - Research and Innovation Framework Programme | GA-965286 | José Jalife |
| La Caixa Banking Foundation | LCF/PR/HR19/52160013 | José Jalife |
| Dr. Bernard Lublin | Donation | Ofer Binah |
| Duchenne Parent Project | 2029771 | Ofer Binah |
| National Institute of Arthritis and Musculoskeletal and Skin Diseases | AR068428 | Daniel E Michele |
| US-Israel Binational Science Foundation | 2013032 | Daniel E Michele Ofer Binah |
| US-Israel Binational Science Foundation | 2019039 | Ofer Binah Todd J Herron |

The funders had no role in study design, data collection, and interpretation, or the decision to submit the work for publication.

### Author contributions

Eric N Jimenez-Vazquez, Conceptualization, Investigation, Writing – original draft, Writing – review and editing, Formal analysis, Methodology, Data curation; Michael Arad, Guadalupe Guerrero-Serna, Investigation, Writing – review and editing; Álvaro Macías, Formal analysis, Investigation, Writing – review and editing; Maria L Vera-Pedrosa, Investigation, Resources, differentiation and maturation of iPSC-CMs used in SNTA1 rescue; Francisco Miguel Cruz, Investigation, Methodology, Resources, Writing – review and editing; Lilian K Gutierrez, Formal analysis, Generated all the data presented in Figure 8, Investigation, Methodology, Software; Ashley J Cuttitta, Methodology, Resources, Writing – review and editing; André Monteiro da Rocha, Methodology, Writing – review and editing; Todd J Herron, Daniel E Michele, Methodology, Resources; Daniela Ponce-Balbuena, Investigation; Ofer Binah, Resources, Writing – review and editing; José Jalife, Conceptualization, Data curation, Formal analysis, Funding acquisition, Investigation, Methodology, Project administration, Supervision, Validation, Writing – original draft, Writing – review and editing

### Author ORCIDs

Eric N Jimenez-Vazquez ⓘ http://orcid.org/0000-0001-5134-491X
Álvaro Macías ⓘ http://orcid.org/0000-0002-9952-6947
Maria L Vera-Pedrosa ⓘ http://orcid.org/0000-0001-9861-7284

Francisco Miguel Cruz  http://orcid.org/0000-0001-5330-8613
Lilian K Gutierrez  http://orcid.org/0000-0003-1476-723X
Ashley J Cuttitta  http://orcid.org/0000-0002-2400-4087
André Monteiro da Rocha  http://orcid.org/0000-0003-3324-3773
Daniela Ponce-Balbuena  http://orcid.org/0000-0002-7413-5325
Guadalupe Guerrero-Serna  http://orcid.org/0000-0001-9130-1850
Daniel E Michele  http://orcid.org/0000-0003-4393-4551
José Jalife  http://orcid.org/0000-0003-0080-3500

## Ethics

Informed consent, and consent to publish, was obtained. Specific ethical approval was obtained and guidelines were followed including the name of the institutional review board or ethics committee that approved the study.

## Decision letter and Author response

Decision letter https://doi.org/10.7554/eLife.76576.sa1
Author response https://doi.org/10.7554/eLife.76576.sa2

---

## Additional files

### Supplementary files

• Supplementary file 1. Action potential parameters of iPSC-CMs paced at 1 and 2 Hz.

• Supplementary file 2. Action potential parameters of iPSC-CMs paced at 1 and 2 Hz.

• Supplementary file 3. Action potential parameters of iPSC-CMs at 1 and 2 Hz, Control 1 versus Control 2.

• Supplementary file 4. Biophysical parameters of DMD, and female iPSC-CMs versus Control 1.

• Supplementary file 5. Biophysical parameters of DMD, and female iPSC-CMs versus Control 2.

• Supplementary file 6. Biophysical parameters of iPSC-CMs, Control 1 versus Control 2.

• Supplementary file 7. Primers used in mRNA analysis.

• Transparent reporting form

### Data availability

All data generated or analyzed during this study are included in the manuscript and supporting file; source data files have been provided.

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

# Appendix 1

## Supplemental methods

### Ethics statement

We obtained skin biopsies from 2 hemizygous DMD patients, 1 heterozygous female, and a healthy patient after written informed consent in accordance with the Helsinki Committee for Experiments on Human Subjects at Sheba Medical Center, Ramat Gan, Israel (Approval number: 7603-09-SMC), and with IRB HUM00030934 approved by the University of Michigan Human IRB Committee. The use of iPS cells and iPSC-CMs was approved by the Human Pluripotent Stem Cell Research Oversight (HPSCRO, #1062) Committee of the University of Michigan and by the Spanish National Center for Cardiovascular Research (CNIC) Ethics Committee and the Regional Government of Madrid.

### Generation of iPSCs

Cell lines were generated using Sendai virus CytoTune-iPS 2.0 Sendai reprogramming kit (Thermo Fisher) for transfection of Yamanaka's factors: OCT4, KLF4, c-Myc, and SOX2, as described (*Eisen et al., 2018*; *Eisen et al., 2019*). Subsequently, iPSCs were cultured on Matrigel (Corning)-coated 6-well plates with mTeSR1 medium (Stemcell Technologies) at 37°C with 5% $CO_2$. iPSCs were passaged every 5 days at a ratio of 1:6 by mechanical dissociation using 1 mL/well of Versene solution (Invitrogen) following incubation at 37 °C for 7 min. DMD iPSCs were transported from Israel in dry ice to Michigan and to CNIC where they were differentiated to iPSC-CMs and used for the initial (Michigan) and syntrophin rescue (CNIC) studies. All iPSCs were tested for pluripotency before starting cardiomyocyte differentiation protocols. All of cells correlated well with the expression status of the pluripotency factors. Differentiation markers were also assessed.

### Patient-specific iPSC-CMs monolayers: Differentiation into cardiomyocytes, adapted from *Herron et al., 2016*

iPSC-CMs were generated by directed differentiation, modulating Wnt/β-catenin signaling (*Lian et al., 2013*). Briefly, iPSCs were cultured for 5–6 days on Matrigel-coated (Corning, 100 µg/mL) 6-well plates in StemMACs iPSC Brew XF medium (Miltenyi Biotec). Then, iPSCs were dissociated using 1 mL/well Versene solution at 37°C for 7 min and reseeded as monolayers on Matrigel-coated 12-well plates at a density of $8.5 \times 10^6$ cells/well in StemMACs iPSC Brew XF medium supplemented with 5 µmol/L ROCK inhibitor (Miltenyi); medium was replaced every day. After 2 days, when monolayers reached 100% confluence, the medium was changed to RPMI supplemented with B27 minus insulin (Invitrogen) containing 10 µmol/L CHIR99021; this day was labelled as day 1 of differentiation. On day 2, the medium was changed to RPMI supplemented with B27 minus insulin. On day 4, the medium was changed to RPMI supplemented with B27 minus insulin, containing 10 µmol/L of IWP-4. On day 6, the medium was changed to RPMI supplemented with B27 minus insulin. Finally, from the 8th day onwards, the medium was changed to RPMI supplemented with B27 complete supplement, RPMI +B27 media (Invitrogen).

### Patient-specific iPSC-CMs monolayers: Post directed differentiation iPSC-CMs purification using MACs negative selection

The directed differentiation method used here does not generate a completely pure iPSC-CM population. Hence, the following purification steps preceded any characterization or experiments. iPSC-CMs ≥30 days in culture were washed with DPBS (Gibco) and dissociated using 1 mL of 0.25% Trypsin/EDTA per well. Next, 2 mL of EB20 media was added per well of dissociated cells, each well was triturated and then transferred into a sterile 15-mL conical. The EB20 media was composed of: 80% DMEM/F12 (Gibco), 0.1 mM Non-Essential Amino Acids (Gibco), 1 mM L-Glutamine (Gibco), 0.1 mM β-mercaptoethanol (Gibco), 20% Fetal Bovine Serum (FBS, Corning), and 10 µM Blebbistatin (Toronto Research Chemicals). Collected cells were centrifuged at 900 RPM for 5 min at 4°C. *Purification*: (*Pekkanen-Mattila et al., 2019*; *Herron et al., 2017*) After removal of the supernatant, 6 mL of MACs Buffer was added followed by trituration. Cells were centrifuged again at 900 RPM for 5 min at 4°C. The supernatant was aspirated and 80 µL of MACs Buffer was added to resuspend the pellet. Then, 20 µL of non-cardiomyocyte depletion cocktail (-Biotin conjugated) primary antibody was added, flicked 5 times to mix and incubated on ice for 5 min. After primary antibody incubation, 1 mL MACs Buffer was added, and cells were gently triturated followed by a 900 RPM spin for 5 min at 4°C. The excess primary antibody was aspirated and 80 µL of MACs Buffer was used to resuspend the pellet. Next, it was mixed with 20 µL of anti-Biotin magnetic microbeads (secondary antibody)

and incubated on ice for 5 min. In the meantime, LS columns with 30 μm separation filters were placed onto a Quadro MACS Separator magnet, 15-mL conical tubes were appropriately labeled and positioned under each column, and 3 mL of MACs Buffer was run through each column to prime for addition of cell suspension. After secondary antibody incubation, cells were mixed with 1 mL of MACs Buffer. Then, the cell suspension was added to the separating filter on top of the flowing column, followed by 3 × 3 mL of cold MACs Buffer washes while continuously collecting the total flow through. The flow through or iPSC-CMs fraction was triturated and 1 mL of the total suspension was placed in a 1.5 μmL Eppendorf tube to count the iPSC-CMs using a Millipore Scepter with Sensor tips (60 μm), this 1 mL was added back to the iPSC-CMs suspension total. Next, the purified iPSC-CMs were centrifuged, the supernatant aspirated, and then resuspended in media for plating. Plating. The purified (98-99%) iPSC-CMs fractions were resuspended in EB20 media with 5 μM of ROCK inhibitor to 200–300k cells/200–300 μL volume and plated as monolayers on 22 mm × 22 mm cut Matrigel-coated (100 μg/mL diluted in DMEM/F12 media) PDMS. The plate was transferred to the incubator at 37°C and 5% CO2 for 2 hours. Next, 3 mL of EB20/ROCK inhibitor media was added to each well. After 2 days, iPSC-CMs were washed with 3 mL DPBS with $Ca^{2+}$ and $Mg^{2+}$ (Gibco) followed by addition of 3 mL of RPMI +B27 media; media was changed every 3 days. The highly purified iPSC-CMs were in monolayer culture on Matrigel-PDMS for at least 7 days after plating to induce maturation. Then monolayers were dissociated with 0.25% Trypsin/EDTA and re-plated onto Matrigel-coated micropatterned PDMS. All iPSC-CM selection materials were purchased from Miltenyi Biotec, except for culture media which was mixed in the laboratory. All the tests carried out in this study were performed using at least 3 separate cardiomyocyte differentiations.

## Micropatterning on PDMS (adapted from *Kuo et al., 2012*)

Micropatterned area was 1 cm × 1 cm total, each island was 100 μm length × 15 μm width and islands were spaced 80 μm from each other.

### Preparing PDMS stamps

The surface of stamps was cleaned with scotch tape followed by sonication in 70% ethanol/milli-Q water for at least 20 min. In a sterile hood, they were allowed to dry and then, incubated with 250 μL Matrigel (100 μg/mL) diluted in water at room temperature for at least 1 h.

### Preparing PDMS substrates in 6-well plates

18 mm PDMS circles were sonicated in 70% ethanol for 20 min and transferred to a 6-well plate after shaking excess EtOH off. When ready for microprinting, the culture dish was UVO treated with the lid off for 9 min.

### Microprinting

While UVO is performed on PDMS circles, the Matrigel solution from the PDMS stamps was aspirated. After UVO was completed, dried stamps were inverted onto each PDMS circle and removed one by one after ~2 min. Later, the micropatterned PDMS plate was incubated with pluronic-F127 overnight at room temperature.

### Single cell re-plating

Before re-plating iPSC-CMs, micropattern plates were cleaned with 3× PSA (Penicillin-Streptomycin-Amphotericin B solution; Thermo Scientific) diluted in PBS (Gibco) for 1 h, and exposed to UV light for 15 min. iPSC-CMs were dissociated from monolayers using trypsin 0.25% with EDTA for 8–10 min and adding RPMI media containing 10% FBS after dissociation. Next, dissociated iPSC-CMs were transferred through a 70 μm filter into a 50-mL conical tube. The iPSC-CM suspension was centrifuged at 700 RPM for 3 min. Subsequently, iPSC-CMs were re-suspended in warm RPMI/B27+ (with insulin) media supplemented with 2% FBS and 5 μM ROCK inhibitor (re-plating media). Finally, ~30k iPSC-CMs in 350 μL re-plating media were placed in the center of the micropatterned area. After ~5 h, 2 mL of re-plating media was added very gently. Plate was returned to the incubator and media change was performed at days 1 and 3 after re-plating. iPSC-CMs were on micropatterns at least 4 days prior to patch-clamping experiments.

## Electrophysiology

Standard patch-clamp recording techniques were used to measure action potentials, $I_{Na}$, $I_{CaL}$, and $I_{K1}$ (**Herron et al., 2016**; **Caballero et al., 2017**). All the experiments were performed at room temperature (22°C–25°C), except for the AP that were recorded at 37°C.

Voltage-clamp experiments were controlled with a Multiclamp 700B amplifier and a Digidata 1440A acquisition system (Molecular Devices). Data were filtered at 5 kHz and sampled at 5–20 kHz. Activation curve data were fitted to a Boltzmann equation, of the form g = $g_{max}$ / (1 + exp ($V_{50}$ − $V_m$) / k), where g is the conductance, $g_{max}$ the maximum conductance, $V_m$ is the membrane potential, $V_{50}$ is the voltage at which half of the channels are activated, and k is the slope factor.

Pipettes were formed from aluminosilicate glass (AF150-100-10; Science Products) with a P-97 horizontal puller (Sutter Instruments), and had resistances between 2 and 3 MΩ for patch-clamp experiments and 5–7 MΩ for current-clamp recordings when filled with the respective pipette solutions (see below).

## Action potential recordings

APs were elicited at 1 and 2 Hz in current-clamp mode using a programmable digital stimulator. The iPSC-CMs were bathed in 148 mM NaCl, 0.4 mM $NaH_2PO_4$, 1 mM $MgCl_2$, 5.4 mM KCl, 1.8 mM $CaCl_2$, 15 mM HEPES, and 5.5 mM glucose, pH = 7.4 adjusted with NaOH. The pipette solution contained 150 mM KCl, 1 mM $MgCl_2$, 1 mM EGTA, 5 mM HEPES, 5 mM phosphocreatine, 4.4 mM $K_2ATP$, and 2 mM β-hydroxybutyric acid, pH = 7.2 adjusted with KOH. Action potential properties including, maximum diastolic potential, overshoot, action potential amplitude and action potential duration were analyzed using custom-made software developed by Krzysztof Grzeda for the Center of Arrhythmia Research, University of Michigan. Maximum upstroke velocity was estimated using OriginPro 9 (OriginLab Corporation). For current clamp experiments, we selected control iPSC-CMs with ventricular-like action potentials showing a rectangular configuration, upstroke velocities (dV/dtmax) ≥40 V/s and amplitudes of 100 mV in the controls. Cells with a triangulated action potential (i.e., atrial-like), depolarized MDP and steep phase-4 depolarization (node-like) were discarded. All iPSC-CMs selected for patch clamping were quiescent and required external stimulation to generate action potentials.

## Single currents

For $I_{Na}$, we used a pulse protocol from -80 mV to +55 mV with a holding potential of -160 mV. Recordings were made in a bath solution that consisted of 10 mM NaCl, 1 mM $MgCl_2$, 0.1 mM $CdCl_2$, 20mM HEPES, 11 mM Glucose, 60 mM CsCl, and 72.5 mM Choline chloride, pH = 7.35 adjusted with CsOH. The pipette solution contained 60 mM CsF, 5 mM NaCl, 10 mM EGTA, 5 mM HEPES, 5 mM MgATP, and 75 mM Choline chloride, pH = 7.2 adjusted with CsOH.

$I_{K1}$ was elicited from a holding potential of -50 mV by 500-ms steps from -120 to +40 mV. The external recording solution contained 148 mM NaCl, 0.4 mM $NaH_2PO4$, 1 mM $MgCl_2$, 5.5 mM Glucose, 1.8 mM $CaCl_2$, 5.4 mM KCl, 15 mM HEPES, and 5 μM Nifedipine, pH = 7.4 adjusted with NaOH. 1 mM $BaCl_2$ was used to isolate $I_{K1}$ from other background currents (subtract solution). The internal solution contained 1 mM $MgCl_2$, 5 mM EGTA, 140 mM KCl, 5 mM HEPES, 5 mM Phosphocreatine, 4.4 mM $K_2ATP$, and 2 mM β-Hydroxybutyric acid, pH = 7.2 adjusted with KOH.

$I_{CaL}$ was evoked applying a voltage-step protocol from -40 mV to +80 mV with a holding potential of -50 mV. The iPSC-CMs were bathed in 137 mM TEA-Cl, 5.4 mM CsCl, 1 mM MgCl2, 1.8 mM CaCl2, 4 mM Aminopyridine, 10 mM HEPES, 30 μM TTX, and 11 mM Glucose, pH = 7.4 adjusted with CsOH. The pipette solution contained 20 mM TEA-Cl, 120 mM CsCl, 1 mM MgCl2•6H2O, 5.2 mM Mg-ATP, 10 mM HEPES, and 10 mM EGTA, pH = 7.2 adjusted with CsOH.

Chemicals were purchased from Sigma. Data analysis was performed using pClamp 10.2 software package (Axon Instruments).

## RT-PCR

For quantitative evaluation of the steady-state mRNA expression in iPSC-CM cultures, total RNA was prepared using the RNeasy Mini Kit (Qiagen), including DNAse treatment. 300 ng of RNA were reversed transcribed and converted to cDNA with oligo(dT)15 primers using reverse transcriptase according to manufacturer's specifications, SuperScript III First-Strand Synthesis System (Invitrogen). Quantitative PCR was performed using Sybergreen Master Mix (Applied Biosystems) in the presence of sense- and antisense-primers (10 μM) for SCN5A, CACNA1C and KCNJ2, as described previously

(*Bizy et al., 2013*). The PCR condition consisted of 95°C for 5 min, followed by 40 cycles of 95°C for 15 secs and 60°C for 1 min, followed by melting-curve analysis to verify the correctness of the amplicon.

The samples were analyzed in biological triplicates using the primers listed in supplemental table 1 and run in a StepOnePlus Real-Time PCR system (Applied Biosystems). The expression of the mRNA of the gene of interest relative to the internal control 18s rRNA in samples from control, hemizygous and heterozygous iPSC-CMs was calculated by the ΔΔCT method, based on the threshold cycle (CT), as fold change = $2^{-(\Delta\Delta CT)}$, where $\Delta CT = CT_{gene\ of\ interest} - CT_{18S}$ and $\Delta\Delta CT = \Delta CT_{hemizygous/heterozygous\ iPSC\text{-}CMs} - \Delta CT_{control\ iPSC\text{-}CMs}$ (*Schmittgen and Livak, 2008*). From each experiment, the cDNA of 3 cell culture wells were measured as biological replicates of each cell line. Each cell culture well was measured from at least 3 separate cardiomyocyte differentiation cultures as technical replicates.

## Western blotting: Cell surface protein biotinylation/Western Blot

iPSC-CMs were plated as above, and membrane proteins were biotinylated. iPSC-CMs monolayers were washed twice with ice cold PBS and biotinylated for 1 h at 4°C using PBS containing 1.5 mg of EZ Link Sulfo-NHS-SS-Biotin (Thermo Scientific). Next, each monolayer was washed 3× with PBS before and after 10 min/4°C incubation with PBS/100 mM Glycine (to quench unlinked biotin). Finally, iPSC-CMs were lysed for 1 h at 4°C with lysis buffer containing (in mM, pH = 7.4): 150 NaCl, 25 Tris, 1% Triton X, and 1% Sodium deoxycholate, supplemented with protease inhibitors consisting of 1 µg/mL Benzamidine, 2 µg/mL Leupeptin, and 2 µg/mL Pepstatin A.

## Western blotting: Protein precipitation

Pull-down experiments were conducted overnight at 4°C with 30 µg of biotinylated protein dissolved in 100 µL of lysis buffer and 30 µL of Pierce Streptavidin magnetic beads (Thermo Scientific). Next day, magnetic beads were washed three times with lysis buffer, and the first supernatant was collected. 25 µL of 4× loading buffer were then added to the magnetic beads. Before loading samples into the gel, they were heated at 50°C for 5 min.

## SDS/PAGE and immunoblotting

Proteins were resolved in 4–20% SDS-PAGE gels and transferred to iBlot stacks with regular PVDF membranes using the Life Technologies iBlot2 system. Nonspecific binding sites were blocked with 5% albumin in PBS-T (in mM, 3 KH2PO4, 10 Na2HPO4, 150 NaCl, and 0.1% Tween 20, pH = 7.2–7.4) for 30 min at room temperature. Membranes were probed with the anti-human NaV1.5 or Kir2.1 antibody diluted in 5% albumin/PBS-T overnight at 4°C. After washing 3×/10 min, membranes were incubated for 1 h with a secondary horseradish peroxidase-conjugated antibody diluted in 5% albumin/PBS-T. Subsequently, membranes were washed 3×/10 min with PBS-T. Signals were detected with the SuperSignal West Pico Chemiluminescent substrate (Thermo Scientific). Expression of NaV1.5 and Kir2.1 was quantified using Image Lab software (Bio-Rad).

Primary antibodies were prepared in block solution. Mouse anti-Cardiac Troponin T antibody (1:1000, #Ab10214, Abcam) was used to identify cTnT as the marker for cardiomyocytes. Rabbit anti-NaV1.5 antibody (clone ASC-013, Alomone Labs) was used for NaV1.5 protein expression (1:500), mouse anti-Kir2.1 antibody (clone N112B/14, University of California at Davis/Nacional Institutes of Health 105 NeuroMab Facility) was used for Kir2.1 protein expression (1:500), mouse anti-Dystrophin (1:000, #D8043, Sigma) was used to detect the Dp427 dystrophin isoform. Mouse anti-Actinin antibody (1:1000, #A7811, Sigma) was used to detect Actinin, loading control in total protein analysis. A mouse antibody (#Ab7671, Abcam) was used to detect the Na-KATPase, positive control for biotinylation assays. Rabbit anti-Connexin antibody (1:1000, #C6219, Sigma) was used to detect Connexin 43. HRP-conjugated secondary antibodies (mouse HRP #115-035-146 and rabbit HRP #111-035-144) were obtained from Jackson ImmunoResearch Laboratories for Western blot analysis.

## Immunofluorescence

iPSC-CMs were seeded on micropatterned Matrigel-coated 6-well plates and fixed with 2% paraformaldehyde/PBS for 15 min. Then, iPSC-CMs were washed 5 min with PBS and blocked with block solution (PBS + 5% BSA + 0.4% Triton X) for 1 h. Incubation with primary antibodies was done in block solution for 1.5 h in a humidity chamber. To washout the excess of primary antibody, iPSC-

CMs were washed 3×/5min with PBS. Next, secondary antibodies in block solution were added to each slip and incubated for 1 h in a humidity chamber at room temperature. iPSC-CMs were kept in dark, washed with PBS 3×/5 min, and mounted with PermaFluor Aqueous (Thermo Fisher) and coverslip.

Primary antibodies were used at different dilutions in block solution: Troponin I (#MAB1691, Millipore) was used at 1:500, Dystrophin MANDRA1 (#D8043, Sigma) was used at 1:100, and Phalloidin-488 (#A12379, Invitrogen) at 1:500 (it comes with a fluorophore conjugated so no secondary Ab incubation was needed, stains F-actin). Secondary Ab for cTnI was Cy3 Goat anti-Mouse IgG (1:400, #115-167-003, Jackson Immuno Research), and for anti-dystrophin MANDRA1, the Cy3 Rat anti-mouse IgG (1:200, #415-165-166, Jackson ImmunoResearch) was used. Both secondary Abs were diluted in block solution containing 1:10,000 DAPI (#D9542, Sigma) stain dilution. Immunostained preparations were analyzed by confocal microscopy, using a Nikon A1R confocal microscope 102 (Nikon Instruments Inc) and Leica SP8 confocal microscope (Leica Microsystems) to determine protein localization.

## Optical Mapping

iPSC-CMs were plated as monolayers at a density of ~50k iPSC-CMs in RPMI/B27+ media. After 7 days in culture, media was removed and each iPSC-CMs monolayer was washed with Hank's balanced salt solution with Ca2+ and Mg2+ added (HBSS++, Thermo Scientific) to remove remaining media. Next, iPSC-CMs were incubated with the FluoVolt membrane potential probe (F10488; Thermo Scientific) diluted in HBSS++, as reported before (*da Rocha et al., 2017*). After a 30-minute incubation time, iPSC-CMs were washed with HBSS++ and then heated at 35°C before optical mapping recordings. All iPSC-CMs monolayers displayed pacemaker activity, and the spontaneous and paced APs were recorded using a charge-coupled device camera (200 fps, 80 × 80 pixels; Red-Shirt Little Joe) with the appropriate emission filters and light-emitting diode illumination (*Lee et al., 2011*). The recorded videos were filtered in both the time and the space domain, and CV was measured as described previously (*Herron et al., 2016*; *Campbell et al., 2012*).

## Generation and Stable Transfection of *SNTA1-IRES-GFP* using PiggyBac Transposon Integration Methods

Non-viral piggy-bac vector (1 μg) encoding SNTA1-IRES-GFP were co-transfected with mouse transposase-expression vector (250 ng) by electroporation (Amaxa 4D-Nucleofector, Lonza) into iPSCs cells (~1.10$^6$ cells/electroporation). After 3-5 days GFP positive cells were selected by FACS sorter (BD FACSAria Cell Sorter, BD BioSciences) and grow-up. Every week, until three times, fluorescence was confirmed, and cells sorted to confirm cDNA stable integration into the cells. After that, iPSC-CMs differentiation protocol was applied as stated above.

## Statistics

Statistical analyses were performed with Prism 8 (GraphPad Software). Values were first tested for normality (Shapiro-Wilk test) before statistical evaluation. Nonparametric Mann-Whitney rank test (two-tailed) was used. Multiple comparisons were analyzed using two-way analysis of variance (ANOVA) followed by Sidak's test. All data are shown as mean± s.e.m. $P < 0.05$ (2-tailed) was considered significant. Unless stated otherwise, the number n of observations indicated reflects the number of iPSC-CMs recorded from each cell line from at least 3 differentiations.

