## [Editor Report]

This paper provides a novel correlation of cardiac with skeletal muscle phenotypes in muscle dystrophic disease. Pluripotent stem cells reprogrammed from cardiomyopathic Duchenne muscle dystrophic patients showed a dysfunctional Na^+^ channel/K^+^ channel channelosome. This correlates with a reduced cardiac excitability and conduction.

---

## [Decision Letter]

**Decision letter after peer review:**

Thank you for submitting your article "SNTA1 GeneRescues Ion Channel Function in Cardiomyocytes Derived from Induced Pluripotent Stem Cells Reprogrammed from Muscular Dystrophy Patients with Arrhythmias" for consideration by *eLife*. Your article has been reviewed by 3 peer reviewers, including Christopher L-H Huang as Reviewing Editor and Reviewer #1, and the evaluation has been overseen by a Reviewing Editor and a Senior Editor.

All three reviewers were very complementary about your work, but two of the reviewers have specific queries, and in one case, a request for some additional data and clarification. The reviewers have discussed their reviews with one another, and the Reviewing Editor has drafted this letter to help you prepare a revised submission.

Essential revisions:

Requests from reviewer #2:

1. SNTA1 appears to restore membrane-localization of Nav1.5/Kir2.1 and partially restore AP, INa and Ik1 flow, and other EP parameters. How is contractility and arrhythmia in rescued DMD-iCM? Arrythmia is mentioned in the title but is not shown in the data?

2. Tied to the above point, would be also helpful to see contractility (eg. shortening) comparison before and after SNTA1 rescue in DMD-iCMs.

3. It is interesting that heterozygote female presents inconclusive result, possibly due to the scarcity of sample. If possible, would like to see data from a different heterozygote individual.

Requests from reviewer #3:

Several suggestions are provided below to improve the rigor of the strong results and clarify some of the findings.

1. It is not clear why the patient with Becker MD is included in Figure 2. However now that it is included it should be pointed out the band is quite faint in Panel 2a and, given the amount of cTnT in that lane, the relative intensity is extremely low. Why is the Becker patient not included in Figure 2b?

2. iPS-CMs from the female patient had the highest (dV/dt) max among the positive patients, yet the lowest CV (Figure 4b) and lowest Ina,peak (Figure 5b). This result is counterintuitive.

3. Despite the lowest Ina,peak, iPS-CMs from the female patient do not appear to have a reduction in IK1 (Figure 5). What are the implications of this result for the Nav1.5-Kir2.1 channelosome?

4. It would be ideal if the elegant rescue experiments in Figures 6 and 7 could be repeated in iPS-CMs from the other two patients as well.

*Reviewer #1 (Recommendations for the authors):*

I felt this was a well written paper. The electrophysiological aspects are thoroughly dealt with and discussed, and so I do not have suggestions for change.

*Reviewer #2 (Recommendations for the authors):*

In the manuscript "SNTA1 Gene Rescues Ion Channel Function in iPSC-CMs from Muscular Dystrophy Patients with Cardiomyopathy and Arrhythmias", authors used DMD donor iPSC-CM to demonstrate electrophysiological dysfunction, specifically pinpointed to the dislocated Nav1.5-Kir2.1 complex, which can be rescued by scaffolding protein a-syntrophin (SNTA1). This model illustrates a very straightforward mechanism of DMD associate cardiomyopathy and offers a feasible/potential treatment. The iPSC in vitro differentiated CM often present with concerns of not resembling rod-shape mature myocytes. This manuscript did an elegant assay by using matrigel-coated PDMS system, yielding morphologically matured myocytes. Beautiful work.

1. SNTA1 appears to restore membrane-localization of Nav1.5/Kir2.1 and partially restore AP, INa and Ik1 flow, and other EP parameters. How is contractility and arrhythmia in rescued DMD-iCM? Arrythmia is mentioned in the title but is not shown in the data?

2. Tied to the above point, would be also helpful to see contractility (eg. shortening) comparison before and after SNTA1 rescue in DMD-iCMs.

3. It is interesting that heterozygote female presents inconclusive result, possibly due to the scarcity of sample. If possible, would like to see data from a different heterozygote individual.

*Reviewer #3 (Recommendations for the authors):*

The manuscript has an important mechanistic conclusion. Several suggestions are provided below to improve the rigor of the strong results and clarify some of the findings.

1. It is not clear why the patient with Becker MD is included in Figure 2. However now that it is included it should be pointed out the band is quite faint in Panel 2a and, given the amount of cTnT in that lane, the relative intensity is extremely low. Why is the Becker patient not included in Figure 2b?

2. iPS-CMs from the female patient had the highest (dV/dt)max among the positive patients, yet the lowest CV (Figure 4b) and lowest Ina,peak (Figure 5b). This result is counterintuitive.

3. Despite the lowest Ina,peak, iPS-CMs from the female patient do not appear to have a reduction in IK1 (Figure 5). What are the implications of this result for the Nav1.5-Kir2.1 channelosome?

4. It would be ideal if the elegant rescue experiments in Figures 6 and 7 could be repeated in iPS-CMs from the other two patients as well.

---

## [Author Response]

Reviewer #1 (Recommendations for the authors):I felt this was a well written paper. The electrophysiological aspects are thoroughly dealt with and discussed, and so I do not have suggestions for change.

We thank Dr. Christopher L-H Huang who served as Reviewer 1, for actively participating in the review process and for his very positive comments about our manuscript.

Reviewer #2 (Recommendations for the authors):In the manuscript "SNTA1 Gene Rescues Ion Channel Function in iPSC-CMs from Muscular Dystrophy Patients with Cardiomyopathy and Arrhythmias", authors used DMD donor iPSC-CM to demonstrate electrophysiological dysfunction, specifically pinpointed to the dislocated Nav1.5-Kir2.1 complex, which can be rescued by scaffolding protein a-syntrophin (SNTA1). This model illustrates a very straightforward mechanism of DMD associate cardiomyopathy and offers a feasible/potential treatment. The iPSC in vitro differentiated CM often present with concerns of not resembling rod-shape mature myocytes. This manuscript did an elegant assay by using matrigel-coated PDMS system, yielding morphologically matured myocytes. Beautiful work.

We also thank this reviewer for recognizing the quality and relevance of our study.

1. SNTA1 appears to restore membrane-localization of Nav1.5/Kir2.1 and partially restore AP, INa and Ik1 flow, and other EP parameters. How is contractility and arrhythmia in rescued DMD-iCM? Arrythmia is mentioned in the title but is not shown in the data?

These are very important and relevant questions which have led us to conduct new optical mapping experiments. To address the issue of lack of arrhythmia data, we determined whether by rescuing the defective Na_V_1.5-Kir2.1 channelosome, *SNTA1* gene expression also prevents arrhythmias. These data are presented as new Figure 8 demonstrating that SNTA1 improves conduction velocity toward control (Figure 8a and b), shortens the optical action potential duration. Most important, as demonstrated by the data presented in Figure 8e, as well as in the supplemental movies 2-5, lack of dystrophin in the Male 1 iPSC-CM monolayers significantly increases the inducibility of reentrant arrhythmias (rotors) yielding very complex and highly variable patterns of nonlinear wave propagation throughout the monolayer, recapitulating episodes of ventricular flutter and ventricular fibrillation recorded by ECG in this patient (Figure 1b). As summarized in Figure 8f, *SNTA1* transfection eliminated arrhythmia inducibility in 9/10 monolayers from this patient. These data fully justify the mention of arrhythmia in the title.

2. Tied to the above point, would be also helpful to see contractility (eg. shortening) comparison before and after SNTA1 rescue in DMD-iCMs.

As recommended, we have conducted additional experiments to measure contractility (i.e., cell shortening) in single iPSC-CMs a well as iPSC-CM monolayers as illustrated in new *Figure 8 —figure supplement 1*. We demonstrate that iPSC-CMs from Male 1 have a significant defect in contractility, manifested as prolonged shortening (corroborated as a reduced slope) and relaxation, and a reduced amplitude of shortening. These data highlight the importance of Na_V_1.5-Kir2.1 channelosome function in helping to ensure a rapid action potential upstroke leading to an adequate excitation-contraction coupling and contraction.

3. It is interesting that heterozygote female presents inconclusive result, possibly due to the scarcity of sample. If possible, would like to see data from a different heterozygote individual.

We agree with the reviewer that the iPSC-CM data derived from the female patient are not as conclusive as we have liked. The results were unexpected even to us, and we recognize that the small number of patients may be a limitation. However, as discussed in the manuscript, one possible explanation is that heterozygous females may have a mosaic of two or more cell types with dramatically different levels of dystrophin expression due to random inactivation of one of the X chromosomes. As such, females with DMD might be more prone to arrhythmias due to spatial electrical inhomogeneity caused by variable expression of the mutant allele. Variable dystrophin expression in females most likely results in variable Na*_V_*1.5 protein levels, whereas Kir2.1 expression and function are positively modulated to aid trafficking of the few pools of Na_V_1.5 channels belonging to the remaining DAPC, or possibly another pool belonging to a different protein-complex. Nonetheless, the significantly reduced I_Na_ and slowed CV we have observed in the cells from the heterozygous patients clearly correlate with her clinical data. Increasing the sample size was also a part of our initial plan, and it was clear that it would have a significant impact on the conclusions about the defects in cardiac electrical activity in female carriers. Unfortunately, obtaining samples from relatives or other female DMD carriers was no possible. Therefore, we have included this point in the Limitations section of the manuscript acknowledging that our hypothesis regarding X chromosome silencing should be validated in other heterozygote females and/or by deeper genomic analysis.

Reviewer #3 (Recommendations for the authors):The manuscript has an important mechanistic conclusion. Several suggestions are provided below to improve the rigor of the strong results and clarify some of the findings.

We appreciate reviewer 3's insightful comments and suggestions for improving the impact and clarity of our work for all readers.

1. It is not clear why the patient with Becker MD is included in Figure 2. However now that it is included it should be pointed out the band is quite faint in Panel 2a and, given the amount of cTnT in that lane, the relative intensity is extremely low. Why is the Becker patient not included in Figure 2b?

We fully agree. However, for this manuscript, we only included samples from human tissues to which we had access as a visual example of the reduction in dystrophin levels obtained from ventricular tissue of patients as a reference to compare with. As we acknowledge in the legend of Figure 2, the protein used to generate these data was generously provided by the Hypertrophic Cardiomyopathy Clinic, University of Michigan. Although the original plan was to enroll a Becker MD patient, unfortunately this was not possible in the end. The dystrophin band corresponding to the left ventricle of this patient is faint because Baker MD patients do express partially truncated dystrophins, which may retain specific functional properties of full-length dystrophin. Thus, while cTnT is highly expressed in Becker MD tissues, dystrophin is not expressed at the same level in dystrophic ventricular tissue as in healthy patients, despite being more highly expressed than in DMD patients.

2. iPS-CMs from the female patient had the highest (dV/dt)max among the positive patients, yet the lowest CV (Figure 4b) and lowest Ina,peak (Figure 5b). This result is counterintuitive.

We appreciate the reviewer´s insight. These data are intriguing, and finding an explanation has proven difficult. However, we would like to highlight that even though the tendency is real, dV/dt values are close (8±1 for Male 2; 11±1 for Male 1; and 12±2 for Female) and not statistically different among all three patients (please refer to Supplementary Table 1). On the other hand, slowing of CV is known to occur through a variety of mechanisms, including decreased sodium channel availability, and decreased gap junction coupling. We did not observe changes in Cx43 expression. However, it is important to note that the dV/dt_max_ of each individual cell in a monolayer depends not only on the intrinsic dV/dt_max_ but also on the resting potential, the dV/dt of, and the electrical load provided by neighboring cells through the gap junctions. As such the chimeric nature of the dystrophin mutation in the heterozygous female monolayers and the consequent heterogeneity of proteins expression (as commented above) may have resulted in a reduced overall conduction velocity despite the high dV/dt measured by patch clamping in the isolated iPSC-CMs.

3. Despite the lowest Ina,peak, iPS-CMs from the female patient do not appear to have a reduction in IK1 (Figure 5). What are the implications of this result for the Nav1.5-Kir2.1 channelosome?

This excellent question is related the previous one. We know that different pools of Na_V_1.5 and Kir2.1 channels exist that are completely independent of DAPC integrity (PMID: 30232268). As a result of the variable expression of dystrophin in female carriers, the levels of Na_V_1.5/Kir2.1 protein vary. In our case, it is likely that the expression and function of the unaffected Kir2.1 channels positively modulates trafficking of the few pools of Na_V_1.5 channels belonging to the remaining somewhat functional DAPC. As a result, depending on the chimeric nature of the female carrier cells, the expression and function of the Na_V_1.5-Kir2.1 in regulating cardiac electrical activity will vary.

4. It would be ideal if the elegant rescue experiments in Figures 6 and 7 could be repeated in iPS-CMs from the other two patients as well.

We fully agree. Unfortunately, we no longer have access to cells from those two patients as one has died, and the other is not available to provide additional somatic cells for iPSC-CM derivation.